# Are Uncertainty Quantification Capabilities of Evidential Deep Learning a Mirage?

**Maohao Shen**[1][*], **J. Jon Ryu**[1][*], **Soumya Ghosh**[2][†],
**Yuheng Bu**[3], **Prasanna Sattigeri**[2], **Subhro Das**[2], **Gregory W. Wornell**[1]

[1]Department of EECS, MIT, Cambridge, MA 02139
[2]MIT-IBM Watson AI Lab, IBM Research, Cambridge, MA 02142
[3]Department of ECE, University of Florida, Gainesville, FL 32611

{maohao,jongha,gww}@mit.edu,
{ghoshoso,prasanna}@us.ibm.com, subhro.das@ibm.com,
buyuheng@ufl.edu

## Abstract

This paper questions the effectiveness of a modern predictive uncertainty quantification approach, called *evidential deep learning* (EDL), in which a single neural network model is trained to learn a meta distribution over the predictive distribution by minimizing a specific objective function. Despite their perceived strong empirical performance on downstream tasks, a line of recent studies by Bengs et al. identify limitations of the existing methods to conclude their learned epistemic uncertainties are unreliable, e.g., in that they are non-vanishing even with infinite data. Building on and sharpening such analysis, we 1) provide a sharper understanding of the asymptotic behavior of a wide class of EDL methods by unifying various objective functions; 2) reveal that the EDL methods can be better interpreted as an out-of-distribution detection algorithm based on energy-based-models; and 3) conduct extensive ablation studies to better assess their empirical effectiveness with real-world datasets. Through all these analyses, we conclude that even when EDL methods are empirically effective on downstream tasks, this occurs despite their poor uncertainty quantification capabilities. Our investigation suggests that incorporating model uncertainty can help EDL methods faithfully quantify uncertainties and further improve performance on representative downstream tasks, albeit at the cost of additional computational complexity.[1]

## 1 Introduction

Accurate estimation of uncertainty in the prediction becomes more crucial to enhance the reliability of a predictive model, especially for high-stake applications such as medical diagnosis [1, 2]. Among several approaches proposed, a class of uncertainty estimation methods under the category of *evidential deep learning* (EDL) has recently gained attention [3], due to their claimed advantages over other methods. EDL methods typically learn a single neural network that maps input data to the parameters of a meta distribution, which is a distribution over the predictive distribution. The EDL methods generally claim the following advantages. (1) *Computational efficiency*: they bypass the expensive sampling costs associated with Bayesian or ensemble-based methods by training a

---

[*]The two authors contributed equally to this work.
[†]This author is currently affiliated with Merck and can be reached at Soumya.Ghosh@merck.com.
[1]The code to replicate the experiments is available on https://github.com/maohaos2/EDL-Mirage.

single neural network and estimating uncertainty with a single forward pass. (2) *Promising empirical performance*: they achieve superior results on downstream uncertainty quantification (UQ) tasks, particularly in detecting out-of-distribution (OOD) data. (3) *Ability to quantify different uncertainties*: EDL methods can quantify *distributional uncertainty* versus *aleatoric uncertainty*, by representing them as the spread and mean of an estimated meta distribution over the prediction, respectively.

Despite the above benefits, a line of recent works has reported theoretical limitations and pitfalls of uncertainties learned by EDL methods, including the issue of non-vanishing distributional uncertainty [4], a possibility of non-existence of proper scoring rules for meta distributions [5], and a gap between learned uncertainty and an ideal meta distribution [6]. While these works call for the attention of the UQ community for the raised issues, a comprehensive theoretical understanding of the learned uncertainties from this type of UQ model is still lacking, as the existing analyses focus on a restricted subset of objective functions in the literature. Moreover, they did not explain the empirical success of the EDL methods at downstream tasks, such as OOD detection, despite such issues.

In this paper, we provide a simpler and sharper theoretical characterization of what is being learned by representative EDL methods, and re-examine the empirical success of the EDL methods based on the analysis. More concretely, our contributions are threefold:

1. **Theoretical Analysis**: In Sec. 4, we provide a unifying perspective on several representative EDL methods proposed in the literature, establishing an *exact* characterization of the *optimal* meta distribution defined by existing methods. This reveals that existing methods enforce the meta distribution to fit a sample-size-independent target distribution (Theorem 5.1). This analysis covers a wider class of objective functions and modalities, and is sharper than the prior analyses [4, 6].

2. **Empirical Investigation**: In Sec. 5, we further provide empirical evidence to point out the fundamental limitations of the learned uncertainties by EDL methods, and present several findings showing that existing EDL methods are essentially OOD detectors and hence exhibit their pitfalls.

3. **Insights and Solutions**: In Sec. 6, we explain why model uncertainty seems inevitable for faithful UQ and how we can improve the EDL method accordingly. We propose a new model uncertainty based on the idea of bootstrap, and demonstrate that an EDL model can well distill its behavior and achieve superior UQ downstream task performance compared to the existing methods.

## 2   Related Work

We summarize the literature that is closely aligned with the scope of this paper. We refer the reader to Sec. B in Appendix for an overview of classical UQ literature and a recent survey paper [3] for a comprehensive review of EDL methods.

**EDL Methods.** EDL methods, mainly applied in classification settings, utilize a single neural network to model Dirichlet distributions over label distributions, which can be divided into three categories. (1) *OOD-data-dependent methods*: Earlier works such as Forward Prior Network [7], and Reverse Prior Network [8, 9], proposed to train a model to output sharp Dirichlet distribution for in-distribution (ID) data and flat Dirichlet distribution for OOD data. (2) *OOD-data-free methods*: Subsequently, several methods without OOD data were proposed with various training objectives, including the MSE loss with reverse Kullback–Leibler (KL) regularizer [10, 11], the "VI" loss [12, 13, 14], and the "UCE" loss [15]. (3) *Distillation based methods*: are motivated by training a single model to mimic the behavior of classical UQ approaches, including END2 [16] that emulates ensemble method, and S2D [17] that emulates (Gaussian) random dropout method. EDL methods have also been explored for regression problems [18, 19, 20].

**Critiques of EDL Methods.** Recently, several works have raised concerns about the quality of uncertainty learned by EDL methods. Bengs et al. [4] pointed out the learned distributional uncertainty does not vanish even in the asymptotic limit of infinite training samples. Bengs et al. [5] provided further theoretical arguments for why a proper scoring rule for learning the meta distribution might not exist. More recently, Jürgens et al. [6] argues that the learned uncertainty by EDL methods is inconsistent with a reference distribution. In a similar spirit to these critiques, we offer a sharper analysis to characterize the exact behavior of EDL methods. Our analysis can subsume, generalize, and simplify the existing analyses. See Appendix A for a more in-depth review of these works.

# 3 Problem Setting and Preliminaries

In the predictive modeling, we aim to learn the label distribution $p(y|x)$ over $\mathcal{Y}$ using data $\mathcal{D} = \{(x_i, y_i)\}_{i=1}^N$ drawn from an underlying data distribution $p(x)p(y|x)$ over $\mathcal{X} \times \mathcal{Y}$. Here, $\mathcal{Y}$ is a label set; $\mathcal{Y} = [C] := \{1, \ldots, C\}$ for classification for some $C \geq 1$ and $\mathcal{Y} = \mathbb{R}$ for regression. $\Delta^{C-1}$ denotes the probability simplex over $[C]$. In addition to accurately learning the conditional distribution $\eta_y(x) := p(y|x)$, we wish to quantify "uncertainty" of the learned prediction. We focus on classification in this paper, but some of our analyses also apply to certain existing EDL methods for regression [19, 20]. For the sake of clarity, we defer all related discussion on regression and beyond to Appendix F.

**Bayesian and Ensemble-based Approaches.** Different UQ methods define predictive uncertainties based on different sources of randomness. The *Bayesian* approach is arguably the most widely studied UQ approach, in which a parametric classifier $p(y|x, \psi)$ is trained via the Bayesian principle, and inference is performed with the predictive posterior distribution $p(y|x, \mathcal{D}) := \int p(y|x, \psi)p(\psi|\mathcal{D}) \, \mathrm{d}\psi$, where $p(\psi|\mathcal{D})$ is the model posterior distribution, induced by the prior $p(\psi)$ and likelihood $p(\mathcal{D}|\psi)$. Ensemble-based methods assume a different distribution $p(\psi|\mathcal{D})$ to generate random models given data, e.g., training neural networks with different random seeds. As alluded to earlier, both Bayesian and ensemble approaches are computationally demanding due to the intractability of $p(\psi|\mathcal{D})$ and the need for computing the integration over $\psi$. Moreover, $p(y|x, \mathcal{D})$ can capture *aleatoric uncertainty* (or *data uncertainty*), and the amount of spread over $p(y|x, \psi)$ induced by the model uncertainty of $p(\psi|\mathcal{D})$ is regarded as *epistemic uncertainty* (or *knowledge uncertainty*) for its prediction.

**EDL Approach.** The EDL approach further decomposes the predictive posterior distribution as $p(y|x, \psi) = \int p(\pi|x, \psi)p(y|\pi) \, \mathrm{d}\pi$, where $p(\pi|x, \psi)$ is a *meta distribution* (or called *second-order distribution* [4, 5]) over the prediction at $x$, and $p(y|\pi)$ is a fixed likelihood model. For classification, $\pi \in \Delta^{C-1}$ is a probability vector over $C$ classes, $p(y|\pi) = \pi_y$ is the categorical likelihood model, and $p(\pi|x, \psi)$ is a distribution over the simplex $\Delta^{C-1}$. Oftentimes, $p(\pi|x, \psi)$ is chosen as a conjugate prior of the likelihood model $p(y|\pi)$, like the Dirichlet distribution for classification [7, 8, 13, 15, 20], but sometimes not [10, 18, 11]. Given the full decomposition, $p(y|x, \mathcal{D}) = \iint p(y|\pi)p(\pi|x, \psi)p(\psi|\mathcal{D}) \, \mathrm{d}\psi \, \mathrm{d}\pi$, the uncertainty captured in $p(y|\pi)$ is called aleatoric uncertainty, in $p(\pi|x, \psi)$ is called *distributional uncertainty*, a kind of *epistemic uncertainty*. However, EDL methods often assume the best single model $\psi^\star$ learned with data $\mathcal{D}$, without any randomness in $p(\psi|\mathcal{D})$, or formally setting it to be $\delta(\psi - \psi^\star)$ [7, 3]. It then aims to train the meta distribution $p(\pi|x, \psi)$ under certain learning criteria so that it encodes less uncertainty for ID points $x$ and more for OOD points. While this simplification allows its computational efficiency over the classical methods, as we will argue later, no randomness assumed in the model $p(\psi|\mathcal{D})$ lets all the methods in this framework learn spurious distributional uncertainty. Since a single model $\psi$ is assumed, we use a frequentist notation $p_\psi(\pi|x)$ instead of $p(\pi|x, \psi)$ in that context.

# 4 New Taxonomy for EDL Methods

In this section, we propose a new taxonomy to understand different EDL methods (for classification) in a systematic way. Ignoring the choice of base architectures, we identify that the key distinguishing features in EDL methods are (1) the parametric form of the model and (2) the learning criteria. A wide class of EDL methods can be classified with the taxonomy as in Table 1. Several theoretical implications and empirical consequences of the new taxonomy will be investigated in the next section.

*Table 1:* **New taxonomy of representative EDL methods.** $\mathcal{L}(\psi)$ in Eq. (6) subsumes these as special cases.

| Method (name of loss) | likelihood | $D(\cdot, \cdot)$ | prior $\boldsymbol{\alpha}_0$ | $\gamma_{\text{ood}}$ | $\boldsymbol{\alpha}_\psi(x)$ parameterization |
|---|---|---|---|---|---|
| FPriorNet (F-KL loss) [7] | categorical | fwd. KL | $= \mathbb{1}_C$ | $> 0$ | direct |
| RPriorNet (R-KL loss) [8] | categorical | rev. KL | $= \mathbb{1}_C$ | $> 0$ | direct |
| EDL (MSE loss) [10] | Gaussian | rev. KL | $= \mathbb{1}_C$ | $= 0$ | direct |
| Belief Matching (VI loss) [12, 13] | categorical | rev. KL | $\in \mathbb{R}_{>0}^C$ | $= 0$ | direct |
| PostNet (UCE loss) [15] | categorical | rev. KL | $= \mathbb{1}_C$ | $= 0$ | density w/ single flow |
| NatPN (UCE loss) [20] | categorical | rev. KL | $= \mathbb{1}_C$ | $= 0$ | density w/ multiple flows |

**Criterion 1. Parametric Form of Meta Distribution.** For classification, one distinguishing feature is the parametric form of $\boldsymbol{\alpha}_\psi(x)$ in Dirichlet distribution $p_\psi(\boldsymbol{\pi}|x) = \mathsf{Dir}(\boldsymbol{\pi}; \boldsymbol{\alpha}_\psi(x))$. Earlier works [12, 7, 8, 13, 10] typically parameterize $\boldsymbol{\alpha}_\psi(x)$ by a direct output of a neural network, e.g., exponentiated logits; we call this *direct parameterization*. Later, Charpentier et al. [15] brought up a potential issue with the direct parameterization that $\boldsymbol{\alpha}_\psi(x)$ can take arbitrary values on the unseen (i.e., OOD) data points. They proposed a more sophisticated parameterization of the form to explicitly resemble the posterior distribution update of the Dirichlet distribution $\boldsymbol{\alpha}_\psi(x) \leftarrow \boldsymbol{\alpha}_0 + \mathbf{N}_\psi(x)$, where, for $y \in [C]$, $(\mathbf{N}_\psi(x))_y := N\hat{p}(y)p_{\psi_2}(f_{\psi_1}(x)|y)$, with $\hat{p}(y) := N_y/N$, $N_y$ denotes the number of data points with label $y$, $N := \sum_{y \in [C]} N_y$, $x \mapsto f_{\psi_1}(x)$ a feature extractor, and $p_{\psi_2}(z|y)$ a tractable density model such as normalizing flows [21] for each $y \in [C]$. We call this *density parameterization*. In Sec. 5, we carefully examine the effectiveness of density parameterization.

**Criterion 2. Objective Function.** The desired behavior of EDL model is to output sharp $p_\psi(\boldsymbol{\pi}|x)$ if it is confident, and fall back to output *prior* distribution $p(\boldsymbol{\pi})$ if it is uncertain at an unseen data $x$. To achieve this, various objectives have been introduced with different jargon and motivations, e.g.:

1. Prior Networks (PriorNet) [7, 8] aimed to explicitly encourage $p_\psi(\boldsymbol{\pi}|x)$ to be diffused prior $p(\boldsymbol{\pi})$ for OOD data, and a more concentrated Dirichlet distribution $\mathsf{Dir}(\boldsymbol{\pi}; \boldsymbol{\alpha}_0 + \nu \mathbf{e}_y)$ for ID data, with $\nu \gg 1$, $\boldsymbol{\alpha}_0 = \mathbb{1}_C$ and $\mathbf{e}_y$ as the one-hot true label, by minimizing

$$\mathbb{E}_{p(x,y)}[D(\mathsf{Dir}(\boldsymbol{\pi}; \boldsymbol{\alpha}_0 + \nu \mathbf{e}_y), p_\psi(\boldsymbol{\pi}|x)) + \gamma_{\mathsf{ood}}\mathbb{E}_{p_{\mathsf{ood}}(x)}[D(\mathsf{Dir}(\boldsymbol{\pi}; \boldsymbol{\alpha}_0), p_\psi(\boldsymbol{\pi}|x))] \quad (1)$$

for $D(p, q) = D(p \| q)$ (forward KL) in [7], and $D(p, q) = D(q \| p)$ (reverse KL) later in [8]. It is known that PriorNet with forward KL requires an additional auxiliary loss $\log \frac{1}{p_\psi(y|x)}$ to ensure high accuracy, and the reverse-KL version can outperform without such term.

2. The "Evidential Deep Learning" paper [10] proposed the MSE loss with a reverse KL regularizer:

$$\ell_{\mathsf{MSE}}(\psi; x, y) := \mathbb{E}_{p_\psi(\boldsymbol{\pi}|x)}[\|\boldsymbol{\pi} - \mathbf{e}_y\|^2] + \lambda D(p_\psi(\boldsymbol{\pi}|x) \| \mathsf{Dir}(\boldsymbol{\pi}; \boldsymbol{\alpha}_0)). \quad (2)$$

3. Belief Matching [12, 13] proposed VI loss justified by variational inference framework:

$$\ell_{\mathsf{VI}}(\psi; x, y) := \mathbb{E}_{p_\psi(\boldsymbol{\pi}|x)}\left[\log \frac{1}{\pi_y}\right] + \lambda D(p_\psi(\boldsymbol{\pi}|x) \| \mathsf{Dir}(\boldsymbol{\pi}; \boldsymbol{\alpha}_0)). \quad (3)$$

4. Posterior Networks (PostNet [15] and NatPN [20]) proposed the uncertainty-aware cross entropy (UCE) loss, motivating it from a general framework for updating belief distributions [22]:

$$\ell_{\mathsf{UCE}}(\psi; x, y) := \mathbb{E}_{p_\psi(\boldsymbol{\pi}|x)}\left[\log \frac{1}{\pi_y}\right] - \lambda h(p_\psi(\boldsymbol{\pi}|x)). \quad (4)$$

The hyper-parameter $\lambda > 0$ in objectives 2, 3, and 4 balances the first likelihood term which forces $p_\psi(\boldsymbol{\pi}|x)$ to learn the label distribution and the second regularizer term promotes $p_\psi(\boldsymbol{\pi}|x)$ to be close to the prior $p(\boldsymbol{\pi})$. Below, we reveal that the seemingly different objectives 1, 3, 4 are exactly equivalent, while objective Eq. (2) (different *likelihood*) can also be unified in a single framework.

**A Unifying View.** We now provide a unifying view of the fairly wide class of representative objective functions. First, for convenience of analysis, we define the *tempered likelihood*: for $\nu > 0$, define

$$p^{(\nu)}(\boldsymbol{\pi}|y) := \frac{p^{(\nu)}(\boldsymbol{\pi}, y)}{\int p^{(\nu)}(\boldsymbol{\pi}, y) \, \mathrm{d}\boldsymbol{\pi}}, \quad \text{where } p^{(\nu)}(\boldsymbol{\pi}, y) := \frac{p(\boldsymbol{\pi})p^\nu(y|\boldsymbol{\pi})}{\int p(\boldsymbol{\pi}) \sum_y p^\nu(y|\boldsymbol{\pi}) \, \mathrm{d}\boldsymbol{\pi}}. \quad (5)$$

Objectives 1, 3, 4 assume the likelihood model is categorical, i.e., $p(y|\boldsymbol{\pi}) = \pi_y$, by the conjugacy of the Dirichlet distribution for the multinomial distribution, it is easy to check that $p^{(\nu)}(\boldsymbol{\pi}|y) = \mathsf{Dir}(\boldsymbol{\pi}; \boldsymbol{\alpha}_0 + \nu \mathbf{e}_y)$, where $\mathbf{e}_y \in \mathbb{R}^C$ is the one-hot vector activated at $y \in [C]$. Objective 2 used Gaussian likelihood model $p(y|\boldsymbol{\pi}) = \mathcal{N}(\mathbf{e}_y; \boldsymbol{\pi}, \sigma^2 I_C)$, which does not admit a closed form expression for $p^{(\nu)}(\boldsymbol{\pi}|y)$. The prior distribution is usually defined as $p(\boldsymbol{\pi}) = \mathsf{Dir}(\boldsymbol{\pi}; \boldsymbol{\alpha}_0)$ with $\boldsymbol{\alpha}_0 = \mathbb{1}_C$ (all-one vector). Other choices of prior were also proposed to promote some other desired property [9].

We now introduce a *unified objective function*

$$\mathcal{L}(\psi) := \mathbb{E}_{p(x,y)}[D(p^{(\nu)}(\boldsymbol{\pi}|y), p_\psi(\boldsymbol{\pi}|x))] + \gamma_{\mathsf{ood}}\mathbb{E}_{p_{\mathsf{ood}}(x)}[D(p(\boldsymbol{\pi}), p_\psi(\boldsymbol{\pi}|x))] \quad (6)$$

for some divergence function $D(\cdot, \cdot)$, a tempering parameter $\nu > 0$, and an OOD regularization parameter $\gamma_{\mathsf{ood}} \geq 0$ with a distribution $p_{\mathsf{ood}}$ for OOD samples. The following theorem summarizes how this unified objective subsumes several existing proposals. The proof is deferred to Appendix D.

**Theorem 4.1** (Unifying EDL Objectives for Classification). *Let $p(\boldsymbol{\pi}) = \mathsf{Dir}(\boldsymbol{\pi}; \mathbb{1}_C)$.*

1. *Let $p(y|\boldsymbol{\pi}) = \pi_y$ and let $\gamma_{\mathsf{ood}} > 0$. With $D(p,q) = D(p \parallel q)$ (forward KL) and $D(p,q) = D(q \parallel p)$ (reverse KL), $\mathcal{L}(\psi)$ is equivalent to the objective (Eq. (1)) of forward PriorNet (FPrior-Net) [7] and that of reverse PriorNet (RPriorNet) [8], respectively.*

2. *Let $p(y|\boldsymbol{\pi}) = \mathcal{N}(\mathbf{e}_y; \boldsymbol{\pi}, \sigma^2 I_C)$, and let $\gamma_{\mathsf{ood}} = 0$, $\mathcal{L}(\psi)$ is equivalent to the objective $\mathcal{L}_{\mathsf{MSE}}(\psi) := \mathbb{E}_{p(x,y)}[\ell_{\mathsf{MSE}}(\psi; x, y)]$ (Eq. (2)).*

3. *Let $p(y|\boldsymbol{\pi}) = \pi_y$, $\nu = \lambda^{-1}$, and let $\gamma_{\mathsf{ood}} = 0$. With $D(p,q) = D(q \parallel p)$ (reverse KL), $\mathcal{L}(\psi)$ is equivalent to the objective $\mathcal{L}_{\mathsf{VI}}(\psi) := \mathbb{E}_{p(x,y)}[\ell_{\mathsf{VI}}(\psi; x, y)]$ (Eq. (3)).*

4. *Let $p(y|\boldsymbol{\pi}) = \pi_y$, $\nu = \lambda^{-1}$, and let $\gamma_{\mathsf{ood}} = 0$. With $D(p,q) = D(q \parallel p)$ (reverse KL), $\mathcal{L}(\psi)$ is equivalent to the objective $\mathcal{L}_{\mathsf{UCE}}(\psi) := \mathbb{E}_{p(x,y)}[\ell_{\mathsf{UCE}}(\psi; x, y)]$ (Eq. (4)).*

We first remark that the reverse KL divergence captures most of the cases, except the forward KL PriorNet, which is known to be outperformed by PriorNet with reverse KL. Hence, it suffices to focus on understanding the reverse KL objective. Second, the VI loss in Eq. (3) and UCE loss Eq. (4) turn out to be equivalent to the RPriorNet objective, where the tempering parameter $\nu$ and the regularization parameter $\lambda$ is realted to be reciprocal $\lambda = \nu^{-1}$; see Lemma D.1. Overall, this theorem reveals that different motivations, such as VI or Bayesian updating mechanism of belief distributions, were not very significant, and those objectives turn out to be equivalent to the simple divergence matching in Eq. (6). Given this, distinguishing features that better classify different EDL methods are the choices of (1) likelihood model, (2) prior, (3) use of OOD data, and (4) model parametric form, as shown in Table 1. The recent survey [3] also offers a unified view of different objective functions with these features. However, the dichotomy between "PriorNet-type methods" [7, 8] and "PostNet-type methods" [12, 13, 15, 20, 14] in [3] may not effectively contrast different EDL methods compared to our taxonomy.

We acknowledge that there exist other objective functions that might not be covered by this unified view. We defer the discussion on another line of representative work, including the distillation-based methods [16, 17] to Sec. 6. A notable exception that is not subsumed by this unification is FisherEDL [11], which was recently proposed to use a variant of the MSE loss [10] by taking into account the Fisher information of $p_\psi(\boldsymbol{\pi}|x)$; see Appendix F for a discussion. Finally, similar reasoning can be naturally extended to unify different EDL objectives for other tasks such as regression and count data analysis, including [19, 20]. We defer a detailed discussion to Appendix F.

# 5 Rethinking the Success of EDL Methods

In this section, we aim to reveal the secrets of EDL methods' empirical success through a combination of theoretical and empirical findings. As alluded to earlier, our empirical investigation focuses on the reverse-KL type EDL methods, i.e., RPriorNet [8], Belief Matching (BM) [13], PostNet [15], and NatPN [20], which are representative and widely used in the literature.

## 5.1 What Is the "Optimal" Meta Distribution Characterized By The EDL Objectives?

Based on the unification in Eq. (6), we can provide a sharp mathematical characterization of the optimally learned meta distribution for a wide class of EDL objectives. A direct and important consequence of the divergence minimization view is that we can now characterize the "optimal" behavior of the learned meta distribution of a wide class of EDL objectives, provided that the global minimizer is achieved in the nonparametric and population limit. The following theorem is proved in Appendix E.

**Theorem 5.1.** *For any prior $p(\boldsymbol{\pi})$ and likelihood $p(y|\boldsymbol{\pi})$, we have*

$$\min_\psi \mathbb{E}_{p(x,y)}[D(p_\psi(\boldsymbol{\pi}|x) \parallel p_\nu(\boldsymbol{\pi}|y))] \equiv \min_\psi \mathbb{E}_{p(x)}[D(p_\psi(\boldsymbol{\pi}|x) \parallel p^\star(\boldsymbol{\pi}|x))],$$

$$\text{where } p^\star(\boldsymbol{\pi}|x) := \frac{p(\boldsymbol{\pi}) \exp(\nu \mathbb{E}_{p(y|x)}[\log p(y|\boldsymbol{\pi})])}{\int p(\boldsymbol{\pi}) \exp(\nu \mathbb{E}_{p(y|x)}[\log p(y|\boldsymbol{\pi})]) \, \mathrm{d}\boldsymbol{\pi}}. \tag{7}$$

In words, Theorem 5.1 states that when the model meta distribution $p_\psi(\boldsymbol{\pi}|x)$ is trained with the reverse-KL objective, it is forced to fit a *fixed* target meta distribution $p^\star(\boldsymbol{\pi}|x)$.

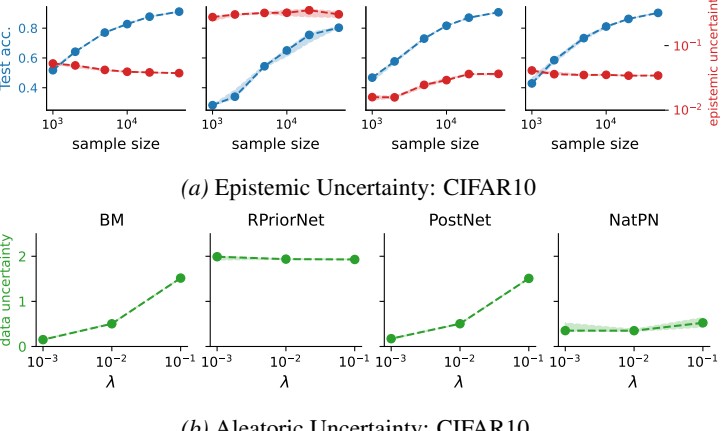

*(a)* Epistemic Uncertainty: CIFAR10

*(b)* Aleatoric Uncertainty: CIFAR10

*Figure 1:* **Behavior of Uncertainties Learned by EDL methods on Real Data.** (a) EDL methods learn spurious epistemic uncertainty, wherein uncertainty does not vanish with an increasing number of observed samples, contrary to the fundamental definition of epistemic uncertainty. (b) Instead of a constant, EDL methods learn model-dependent aleatoric uncertainty that depends on hyper-parameter $\lambda$, contrary to the fundamental definition of aleatoric uncertainty. Similar behavior holds for 2D Gaussian data (see Figure 5 in Appendix H.1).

*Example* 5.2 (Categorical likelihood). If we consider $p(\boldsymbol{\pi}) = \mathrm{Dir}(\boldsymbol{\pi}; \boldsymbol{\alpha}_0)$ and $p(y|\boldsymbol{\pi}) = \pi_y$ (categorical likelihood), we have $p^\star(\boldsymbol{\pi}|x) = \mathrm{Dir}(\boldsymbol{\pi}; \boldsymbol{\alpha}_0 + \nu\boldsymbol{\eta}(x))$, where $\boldsymbol{\eta}(x) := \mathbb{E}_{p(y|x)}[\mathbf{e}_y] = [p(1|x), \ldots, p(C|x)]$ denotes the true label distribution, Theorem 5.1 implies that

$$\min_{\psi} \mathbb{E}_{p(x,y)}[D(p_\psi(\boldsymbol{\pi}|x)\|p_\nu(\boldsymbol{\pi}|y))] \equiv \min_{\psi} \mathbb{E}_{p(x)}[D(\mathrm{Dir}(\boldsymbol{\pi}; \boldsymbol{\alpha}_\psi(x))\|\mathrm{Dir}(\boldsymbol{\pi}; \boldsymbol{\alpha}_0 + \nu\boldsymbol{\eta}(x)))]. \quad (8)$$

In particular, with the most common Dirichlet parameterization $p_\psi(\boldsymbol{\pi}|x) = \mathrm{Dir}(\boldsymbol{\pi}; \boldsymbol{\alpha}_\psi(x))$, this shows that $\boldsymbol{\alpha}_\psi(x)$ is forced to match the scaled-and-shifted version $\boldsymbol{\alpha}_0 + \nu\boldsymbol{\eta}(x)$ of the conditional label distribution $\boldsymbol{\eta}(x)$ as the *fixed* target, under the categorical likelihood model. A similar argument still applies to the Gaussian likelihood model of [10], but the fixed target distribution $p^\star(\boldsymbol{\pi}|x)$ in Eq. (7) does not admit a closed-form expression unlike the categorical likelihood.

An immediate consequence of the Theorem 5.1 is that neither epistemic uncertainty nor aleatoric uncertainty quantified by EDL methods is consistent with their dictionary definition.

**Implication 1: EDL Methods Learn Spurious Epistemic Uncertainty**

First, epistemic uncertainty for ID data, by definition, should monotonically decrease and eventually vanish as the number of observations increases. However, Theorem 5.1 implies that, even with infinite data, the learned "distributional uncertainty" would remain constant for ID data. We also empirically confirm such behavior; see Fig. 1(a). Specifically, we sample data of varying sizes to train the EDL models and evaluate their test accuracy and averaged epistemic uncertainty (mutual information) on a held-out test set. As observed in Fig. 1(a), epistemic uncertainties quantified by EDL methods are almost constant with respect to the sample size and never vanish to 0, regardless of the increasing test accuracy. This suggests that practitioners cannot rely on the learned distributional uncertainty to determine if the model is lacking knowledge. We remark that Bengs et al. [4] identified a similar issue in the EDL methods specifically for the UCE loss of PostNet [15, 20], which is the reverse-KL objective for $\boldsymbol{\alpha} = \mathbb{1}_C$ in our view, is not *appropriate* in a similar spirit.[3] Theorem 5.1 can be understood as a more general and sharper mathematical characterization of the behavior of the reverse-KL objective.

**Implication 2: EDL Methods Learn Spurious Aleatoric Uncertainty**

Second, EDL methods quantify aleatoric (data) uncertainty as $\mathbb{E}_{p_\psi(\boldsymbol{\pi}|x)}[H(p(y|\boldsymbol{\pi}))]$, where $H$ denotes the Shannon entropy [23]. Eq. (8) reveals the optimal meta distribution as $p_{\psi^\star}(\boldsymbol{\pi}|x) =$

---

[3]We remark that the proofs and corresponding arguments in [4] are erroneous, where the errors stem from equating the differential entropy of a Dirac delta function to 0, instead of $-\infty$; see Appendix A for more details.

$\text{Dir}(\boldsymbol{\pi}; \boldsymbol{\alpha}_0 + \nu\boldsymbol{\eta}(x))$, suggesting that the aleatoric uncertainty quantified by EDL methods would also depend on model or algorithm's hyper-parameter ($\lambda = \nu^{-1}$). This is inconsistent with the definition of aleatoric uncertainty, which should be a fixed constant capturing the irreducible uncertainty by underlying label distribution $p(y|x)$. As shown in Fig. 1(b), we empirically demonstrate that the aleatoric uncertainty quantified by EDL methods vary with $\lambda$. In conclusion, our analysis in this section suggests that EDL methods do not behave as a reasonable uncertainty quantifier since they cannot faithfully quantify either epistemic or aleatoric uncertainty.

## 5.2 EDL Methods Are EBM-Based OOD Detector Rather Than Uncertainty Quantifier

As we have shown so far, the EDL methods with Dirichlet prior and categorical model typically aim to fit the model meta distribution $\boldsymbol{\alpha}_\psi(x)$ to a fixed target $\boldsymbol{\alpha}_0 + \nu\boldsymbol{\eta}(x)$, which is approximately $\nu\boldsymbol{\eta}(x)$ for $\nu$ sufficiently large. Since the induced model predictive distribution is $p_\psi(y|x) = \mathbb{E}_{p_\psi(\boldsymbol{\pi}|x)}[p(y|\boldsymbol{\pi})] = \alpha_{\psi,y}(x)/(\mathbb{1}_C^\intercal\boldsymbol{\alpha}_\psi(x))$, it can be understood that the reverse-KL enforces $p_\psi(y|x)$ to fit $\nu\boldsymbol{\eta}(x)/\nu(\mathbb{1}_C^\intercal\boldsymbol{\eta}(x)) = \boldsymbol{\eta}(x) = p(y|x)$ for accurate prediction, while letting the summation $\mathbb{1}^\intercal\boldsymbol{\alpha}_\psi(x)$ large (i.e., $\mathbb{1}^\intercal\boldsymbol{\alpha}_\psi(x) \approx \nu$) for ID data, and small for OOD data, e.g., $\mathbb{1}_C^\intercal\boldsymbol{\alpha}_0 = C$, if there exists an explicit OOD regularization.

In the OOD detection literature, there exists an energy-based-model (EBM) based algorithm aim to achieve the exactly same goal [24]. They consider a standard classifier with exponentiated logits $\boldsymbol{\beta}_\phi(x)$, whose prediction is given as $p_\phi(y|x) := \beta_{\phi,y}(x)/\mathbb{1}_C^\intercal\boldsymbol{\beta}_\phi(x)$. Liu et al. [24] related the denominator to a free energy $E_\phi(x) := -\log\mathbb{1}_C^\intercal\boldsymbol{\beta}_\phi(x)$, and proposed to train model so that $p_\phi(y|x)$ accurately captures $p(y|x)$, while minimizing the energy $E_\phi(x)$ for ID $x$'s, and maximizing for OOD. They proposed to minimize

$$-\mathbb{E}_{p(x,y)}[\log p_\psi(y|x)] + \tau\{\mathbb{E}_{p(x)}[\max(0, E_\phi(x) - m_{\text{id}})^2] + \mathbb{E}_{p_{\text{ood}}(x)}[\max(0, m_{\text{ood}} - E_\phi(x))^2]\},$$

where $\tau > 0$ and $m_{\text{ood}} > m_{\text{id}} > 0$ are hyperparameters. This reveals that this EBM-based OOD framework has an almost identical learning mechanism to the EDL methods; setting $E_\phi(x) = -\log\mathbb{1}_C^\intercal\boldsymbol{\alpha}_\phi(x)$, $m_{\text{id}} = -\log\nu$, and $m_{\text{ood}} = -\log C$ makes the correspondence explicit.

This resemblance suggests that the EDL methods can be better understood as an EBM-based OOD detector with the additional layer of Dirichlet framework for computational convenience rather than a statistically meaningful mechanism that can faithfully distinguish epistemic uncertainty and aleatoric uncertainty. Below, we elaborate on two particular implications based on this connection.

**Implication 3: EDL Methods Always Prefer Smaller Hyper-Parameter $\lambda$**

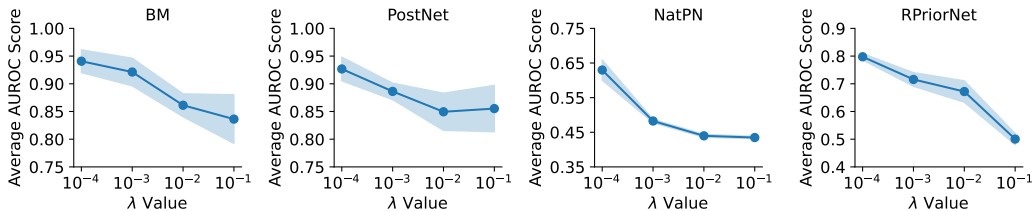

*Figure 2:* **OOD Detection Performance v.s. Hyper-parameter $\lambda$ on CIFAR10.** The $x$-axis represents the increasing $\lambda$ value, and the y-axis represents the Average AUROC score of OOD detection tasks. EDL Methods' uncertainty quantification performance are sensitive to hyper-parameter $\lambda$, while generally benefit from small $\lambda$.

Just as the EBM-based OOD detection algorithm needs to manually define hyper-parameter $m_{\text{ood}}$ and $m_{\text{id}}$, EDL methods also significantly rely on hyper-parameter tuning; recall the target distribution $\text{Dir}(\boldsymbol{\pi}; \boldsymbol{\alpha}_0 + \nu\boldsymbol{\eta}(x))$ with $\nu = \lambda^{-1}$ is determined by the regularization parameter $\lambda$, we aim to explore the dependency of EDL methods' OOD detection performance on the choice of $\lambda$. Specifically, we directly conduct such an analysis on real data using CIFAR10 as ID data, and use four OOD datasets; see Appendix G for details. By varying $\lambda$ across {1e-4,1e-3,1e-2,1e-1}, we train an EDL model and evaluate its OOD detection performance on these OOD datasets. The performance in terms of average AUROC score with respect to the increasing value of $\lambda$ are shown in Fig. 2. The result shows that using smaller values of $\lambda$ always improves the OOD detection performance, suggesting that the EDL model is essentially encouraged to fit its output $\boldsymbol{\alpha}_\psi(x)$ to an extremely large target

$\boldsymbol{\alpha}_0 + \lambda^{-1}\boldsymbol{\eta}(x)$, so that $\mathbb{1}_C^\mathsf{T}\boldsymbol{\alpha}_\psi(x) \approx \lambda^{-1} \gg 1$ for ID data, and such behavior seems to benefit the downstream task performance.

**Implication 4: Impact of Specific Objective on UQ Performance is Less Significant**

Given the close resemblance to the OOD detection algorithm, a notable difference is the reverse-KL learning criterion used by the EDL methods. In Appendix H.2, as another ablation study, we investigate if the reverse-KL objective induced by the Dirichlet framework has a significant practical impact, or other Dirichlet-framework-independent objectives which promote the same behavior suffice for the downstream task performance.

## 5.3 Are EDL Methods Robust for OOD Detection?

Lastly, we investigate the robustness of the auxiliary techniques like density parameterization and OOD regularization of EDL methods for the downstream task performance. Note that in the empirical result with real data presented in Fig. 9 in Appendix, neither OOD regularization (RPriorNet vs. BM) nor density parameterization (PostNet/NatPN vs. BM) demonstrates a clear advantage of deploying such techniques. Thus, we further investigate this counterintuitive phenomenon and discover that the performance of these EDL methods on real data is hindered by certain limitations of the auxiliary techniques they use. Namely, we find that OOD-data-dependent methods [8] are sensitive to choice of model architectures, and methods [15, 20] using density models may not perform well even for moderate dimensionality. We defer the detailed discussion to Appendix H.3.

## 6 EDL Methods Will Benefit from Incorporating Model Uncertainty

Through the series of analyses in the prior section, we attempted to provide a comprehensive understanding of EDL methods, revealing their key limitations. A reader then may ask an important question: **What is the fundamental issue in these EDL methods that cause all these problems?** As we alluded to in Sec. 3, we identify that the issue arises from that all of the EDL methods discussed in Sec. 4 and Sec. 5 assume *no* model uncertainty $p(\psi|\mathcal{D})$ in the decomposition of predictive posterior distribution $p(y|x,\psi) = \iint p(y|\boldsymbol{\pi})p(\boldsymbol{\pi}|x,\psi)p(\psi|\mathcal{D})\,\mathrm{d}\psi\,\mathrm{d}\boldsymbol{\pi}$. A proper definition of distributional uncertainty should be based on the induced posterior distribution $p(\boldsymbol{\pi}|x,\mathcal{D}) := \int p(\boldsymbol{\pi}|x,\psi)p(\psi|\mathcal{D})\,\mathrm{d}\psi$ over the prediction $\boldsymbol{\pi}$ at $x$, given the dataset $\mathcal{D}$. Without the randomness in the model, however, i.e., setting $p(\psi|\mathcal{D}) \leftarrow \delta(\psi - \psi^\star)$, the induced distribution $p(\boldsymbol{\pi}|x,\mathcal{D})$ becomes degenerate as $p(\boldsymbol{\pi}|x,\psi^\star)$. This simplification is often rationalized for the computational efficiency, but as we reveal, it renders the distributional uncertainty inherently ill-defined in this framework. The existing EDL methods thus have no choice but to train the "UQ" model $p_\psi(\boldsymbol{\pi}|x)$ to fit to an artificial target $\mathsf{Dir}(\boldsymbol{\pi}; \boldsymbol{\alpha}_0 + \nu\boldsymbol{\eta}(x))$. Consequently, the learned distributional uncertainty cannot possesses a statistical meaning, and can be better interpreted as free energy in the EBM-based OOD detector.

**Model Uncertainty Can Induce A Proper Distributional Uncertainty.** This strongly suggests that it is inevitable to assume a stochastic procedure $p(\psi|\mathcal{D})$ to properly define the distributional uncertainty $p(\boldsymbol{\pi}|x,\mathcal{D})$. We remark that, to expect the distributional uncertainty $p(\boldsymbol{\pi}|x,\mathcal{D})$ to exhibit a desirable behavior, i.e., getting concentrated for ID data and remaining dispersed for OOD data as $|\mathcal{D}| \to \infty$, we implicitly assume that the stochastic algorithm $p(\psi|\mathcal{D})$ would behave as follows: as $|\mathcal{D}| \to \infty$, $p(\psi|\mathcal{D})$ will become supported on a subset of models $\psi$ that agree upon the prediction for ID data while disagreeing on OOD data. Under this assumption, the induced distributional uncertainty will be consistent with the dictionary definition of epistemic uncertainty.

**Revisiting Distillation-Based Methods.** The downside of such an approach is that approximating $p(\boldsymbol{\pi}|x,\mathcal{D})$ can be computationally intractable due to the high-dimensional integration with respect to the stochastic algorithm $p(\psi|\mathcal{D})$, which means that a practitioner needs to generate (or train) and save multiple models to approximately emulate the randomness. In this context, the EDL framework, which aims to quantify uncertainty by a single neural network, can be used to *distill* the properly defined distributional uncertainty. Indeed, this is what has been explored by another line of EDL literature called *distillation-based methods* [16, 17], which were originally proposed to emulate the behavior the ensemble methods. Our analyses strongly advocate that considering a stochastic algorithm $p(\psi|\mathcal{D})$ and training a single meta distribution trying to fit the induced distributional uncertainty to expedite the inference time complexity is the best practice for the EDL

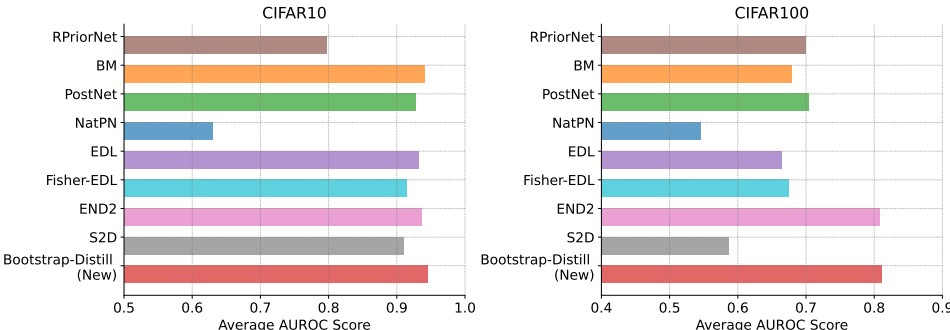

*Figure 3:* **Comparison of Different EDL Methods on OOD Detection.** Distillation based methods, including new proposed Bootstrap-Distill method, demonstrate clear advantage over other classical EDL methods. Similar behavior holds for selective classification task.

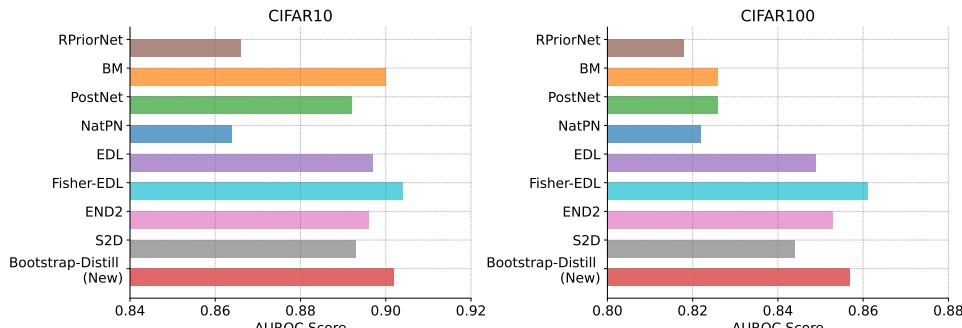

*Figure 4:* **Comparison of Different EDL Methods on Selective Classification.** Distillation based methods, including new proposed Bootstrap-Distill method, demonstrate clear advantage over other classical EDL methods.

framework to faithfully capture uncertainties. More precisely, we can fit an UQ model $p_\theta(\boldsymbol{\pi}|x)$ to $p(\boldsymbol{\pi}|x, \mathcal{D})$ through the forward-KL objective and Monte Carlo samples, i.e., by minimizing $\mathbb{E}_{p(x)}[D(p(\boldsymbol{\pi}|x, \mathcal{D}) \parallel p_\theta(\boldsymbol{\pi}|x))] \approx -\sum_{i=1}^{N} \sum_{j=1}^{M} \log p_\theta(\boldsymbol{\eta}_{\hat{\psi}_j}(\cdot|x_i)|x_i) + (\text{const.})$ with respect to $\theta$, where $x \mapsto \boldsymbol{\eta}_{\hat{\psi}_j}(\cdot|x)$ is a classifier corresponding to a randomly generated model $\hat{\psi}_j \sim p(\psi|\mathcal{D})$ [16].

**Examples of Stochastic Algorithms** $p(\psi|\mathcal{D})$**: Old and New.** There are two popular proposals for $p(\psi|\mathcal{D})$ in the literature: (1) EnD$^2$ [16] considers randomness in random initialization and stochastic optimization, which is called *ensemble* [25], and (2) S2D [17] considers a *random dropout* [26] applied to a single network. We propose yet another algorithm based on the frequentist approach *bootstrap*: given the training dataset $\mathcal{D}$, the procedure randomly samples $M$ different subsets $\{\mathcal{D}_j\}_{j=1}^{M}$ of size $N$ without replacement, and train a model $\hat{\psi}_j$ based on $\mathcal{D}_j$ for each $j$. Unlike the previous proposals of $p(\psi|\mathcal{D})$, the bootstrap method aims to leverage the internal consistency among the ID data, beyond the randomness induced by optimization and architectures. We note that this is inspired by a concurrent work of Jürgens et al. [6], which recently proposed an ideal meta distribution. The proposed bootstrapping can be understood as a practical method for approximating such behavior with finite samples. In the next section, we demonstrate that the new *Bootstrap Distillation* method performs almost best on both OOD detection and selective classification tasks.

## 7 Comprehensive Empirical Evaluation

In this section, we conduct a comprehensive evaluation of the existing EDL methods, including the new proposed Bootstrap Distillation method, on two UQ downstream tasks: (1) *OOD data detection*: identify the OOD data based on the learned epistemic uncertainty; (2) *Selective classification*: identify the wrongly predicted test samples based on total uncertainty, as wrong prediction can occur from either high epistemic or high data uncertainty, or both. These results corroborate several key findings and insights presented in this paper. The detailed experiment setup are provided in Appendix G, and additional experiment results are included in Appendix H.

- **Baselines.** We include a wide range of EDL methods as baselines: Classical methods: (1) RPriorNet [8], (2) Belief Matching (BM) [13], (3) PostNet [15], (4) NatPN [20], (5) EDL [10], and (6) Fisher-EDL [11]. Distillation-based method: (1) EnD$^2$ [16] and (2) S2D [17]. All baseline results are reproduced using their official implementation, if available, and their recommended hyper-parameters.

- **Benchmark.** We consider two ID datasets: CIFAR10, and CIFAR100. For the OOD detection task, we select four OOD datasets for each ID dataset: we use SVHN, FMNIST, TinyImageNet, and corrupted ID data.

- **Evaluation Metric.** We elaborate metrics for quantifying different types of uncertainties in Section C. We evaluate the UQ downstream performance through the Area Under the ROC Curve (AUROC) and Area Under the Precision-Recall Curve (AUPR), where we treat ID (correctly classified) test samples as the negative class and outlier (misclassified) samples as the positive class.

**Results and Takeaway.** The result is summarized in Fig. 3 and Fig. 4. We present the average AUROC score across four OOD datasets for the OOD detection task, and the AUROC score for the selective classification task. More numerical results can be found in Table 2, 3, and 4 in Appendix H. This set of evaluations corroborates our key findings and insights in the previous sections.

Firstly, the results show that varying the likelihood model does not significantly impact performance (BM vs. EDL and Fisher-EDL), supporting the discussion in Sec. 4. Secondly, classical EDL methods achieve comparable performance regardless of the auxiliary techniques, such as density parameterization (PostNet, NatPN), or leveraging OOD data (RPriorNet); this validates our finding in Sec. 5.3. Thirdly, distillation-based methods, particularly EnD$^2$ and the new Bootstrap Distillation, demonstrate superior performance over other baselines, especially on CIFAR100, thereby validating the insights provided in Sec. 6. Moreover, our empirical analysis confirms that the Bootstrap Distillation method can faithfully quantify epistemic uncertainty, as illustrated in Fig. 13 in Appendix H, effectively addressing the limitations identified with other EDL methods in Sec. 5.1. Finally, we note that the performance of Bootstrap Distillation comes at a higher computational cost due to training multiple bootstrap models; see Fig. 14 in Appendix H.

## 8 Concluding Remarks

In this work, we revealed that the uncertainty learned by most of the existing EDL methods bear no statistical meaning. The key theoretical insight is based on the unification of representative EDL objectives in Sec. 5.1. Given this, we suggested that EDL methods can be better interpreted as energy-based OOD detection algorithms, which can explain the reported empirical successes of EDL methods from a different perspective. Additionally, we demonstrated that the performance of EDL methods is sensitive to the choice of auxiliary techniques, further raising questions about their robustness. Finally, we identified that the aforementioned issues with EDL arise from ignoring the model uncertainty for computational efficiency, and argued that the distillation-based methods could potentially remedy the issues, at the cost of additional complexity.

Overall, this work calls researchers' attention to carefully reexamine the capabilities and limitations of the EDL approach at large. For practitioners, EDL methods can still be utilized as efficient algorithms for specific applications, such as OOD data detection. However, when considering EDL methods to build reliable machine learning agents based on their UQ capabilities, practitioners should be aware of their limitations, which contrast with the common belief that EDL approaches can accurately learn and distinguish between epistemic and aleatoric uncertainty.

We conclude the paper with a few remarks on the theory of the Bootstrap-Distill method. As we empirically showed, it can resolve the common issues of the EDL methods with improved downstream task performance, and we thus believe that a careful theoretical analysis of its behavior would be a fruitful direction. While it is relatively easy to argue that the epistemic uncertainty would vanish when the sample size grows to infinity with the bootstrap procedure, we believe that a more sophisticated asymptotic analysis for vanishing epistemic uncertainty could be carried out with overparameterized neural networks, adopting a similar setting in [27]. That is, if the trained model's prediction can be shown to be asymptotically normal in the limit of the sample size, one can argue that the uncertainty captured by bootstrap behaves as Gaussian of vanishing variance in the sample limit. This implies a naturally vanishing epistemic uncertainty. We leave this as a future work.

## Acknowledgments and Disclosure of Funding

The authors appreciate the constructive feedback from anonymous reviewers, which helped improve the manuscript. MS and JJR thank Viktor Bengs for helpful discussions on the literature. This work was supported, in part, by the MIT-IBM Watson AI Lab under Agreement No. W1771646.

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

# Appendix

## A    In-Depth Review of Recent Critiques on EDL Methods

In this section, we conduct a compressive review of the recent critiques of EDL methods [4, 5, 6] that are closely relevant to our paper. We begin by acknowledging the contributions of [4], which pioneered the theoretical study of EDL methods. However, there are several limitations in these works to be noted: (1) Some of the main arguments presented in [4] are invalid given errors in the corresponding proofs. (2) The analysis in these works are purely theoretical. They do not sufficiently explain the empirical behaviors of EDL methods in practical application. (3) The analyses do not include some representative EDL approaches in the literature, such as the distillation-based methods discussed in our paper. (4) While these works effectively point out various limitations of EDL methods, none of these works provide insights or concrete solutions to address these issues.

### A.1    Review of [4] "Pitfalls of Epistemic Uncertainty Quantification Through Loss Minimisation"

In [4], the authors studied the property of a meta distribution characterized by minimizing a certain loss such as the UCE loss [15] for classification. The authors proposed a definition of two desirable properties for a faithfully learned meta distribution should satisfy [4, Definition 1]: First, for some distributional uncertainty measure, the expected uncertainty captured in the meta distribution over observations should monotonically decrease as the data size increases. Second, as the data size goes

to infinity, the meta distribution needs to converge to a degenerate Dirac delta distribution around a single distribution.

They provide three arguments based on the value of regularization parameter: (1) if $\lambda = 0$, which means no regularizer in the UCE loss, then the meta distribution becomes a Dirac function thus the first is violated [4, Theorem 1]; (2) if $\lambda > 0$ in the UCE loss is too large, then the second desideratum is not satisfied [4, Theorem 2]. (3) Also, they claimed that if $\lambda > 0$ is too small, then the first is violated [4, Theorem 3].

We respectfully point out, however, that the proofs of both Theorem 2 and Theorem 3 are erroneous due to the steps that identify the differential entropy of a Dirac delta function as 0, instead of $-\infty$. After correction, we realize the statements of [4, Theorem 3] is no longer valid. Instead, the arguments in [4, Theorem 1] and [4, Theorem 2] can be more sharply inferred from our Theorem 5.1. First, when $\lambda = 0$, it is easy to show that the optimal meta distribution $p^\star(\boldsymbol{\pi}|x)$ in Eq. (7) will be Dirac function. Second, Theorem 5.1 implies a stronger result than the second claim: for *any* value of $\lambda$, the meta distribution converges to a non-Dirac-delta distribution as the sample size grows, thus the second desideratum in [4, Definition 1] is not satisfied. We also note that we do not need an additional regularity assumption.

## A.2 Review of [5] "On Second-Order Scoring Rules for Epistemic Uncertainty Quantification"

As a follow-up work, Bengs et al. [5] studied an existence of a proper scoring rule for second-order distributions (i.e., meta distributions). They extended the standard definition of proper scoring rules [28] to meta distributions, and provided theoretical arguments why there can be virtually no such proper scoring rule, even if we hypothetically assume the existence of the *true* second-order distribution.

At a facial value, this seems to contradict the fact that we use the forward KL objective in the distillation approach. To resolve the inconsistency, we consider the classification case, and we even omit the dependence on the covariate $x$ for simplicity. On the one hand, Bengs et al. [5] considered a particular form of objective in the form

$$L_2(\psi; p_{\mathsf{target}}) := \mathbb{E}_{p_{\mathsf{target}}(\boldsymbol{\pi})}[\mathbb{E}_{y \sim \boldsymbol{\pi}}[\ell(\psi; y)]],$$

for some scoring rule $\ell(\psi; y)$. On the other hand, in the distillation-based method, we use the forward KL objective:

$$L_{\mathsf{fKL}}(\psi; p_{\mathsf{target}}) := D(p_{\mathsf{target}}(\boldsymbol{\pi}) \parallel p_\psi(\boldsymbol{\pi})).$$

Even in the standard EDL methods, we use the reverse-KL objective, though the target may not be meaningful:

$$L_{\mathsf{rKL}}(\psi; p_{\mathsf{target}}) := D(p_\psi(\boldsymbol{\pi}) \parallel p_{\mathsf{target}}(\boldsymbol{\pi})).$$

As the side-by-side comparison reveals, in practice, most existing EDL methods are using either the forward KL objective (distillation based EDL methods) or the reverse-KL objective (classical EDL methods), but the objective function analyzed in [5] assumes a particular form that involves the expectation $\mathbb{E}_{y \sim \boldsymbol{\pi}}[\cdot]$, and this implies that the negative results are of rather limited applicability. As our Sec. 6 suggests, if we can directly construct a reasonable target meta distribution in a way that either KL divergence is computable, then we can use such divergence matching to learn the meta distribution in a faithful manner.

## A.3 Review of [6] "Is Epistemic Uncertainty Faithfully Represented by Evidential Deep Learning Methods?"

In a concurrent work, Jürgens et al. [6] questions the reliability of distributional uncertainty learned by two types of loss under different settings, including classification, regression and count data: the "inner loss" widely used by evidential regression framework [18, 29], and the "outer loss" which is more commonly applied in classification setting and some regression methods [19, 20] (equivalent to general UCE loss in Sec. F in our view). Jürgens et al. [6] mainly aims to examine whether the leared distributional uncertainty can be consistent to a "target" epistemic uncertainty induced by a reference distribution.

First, [6, Definition 3.1] defines such a reference distribution as $q_N(\boldsymbol{\theta}|x) := \int_{(\mathcal{X} \times \mathcal{Y})^N} \mathbb{I}[\boldsymbol{\theta}_{\mathcal{D}_N} = \boldsymbol{\theta}] \, \mathrm{d}P^N$, assuming an access to the underlying data generating distribution $\mathcal{D}_N \sim P^N$ is available,

with $\boldsymbol{\theta}$ representing the true first-order target for input $x$. The term $\boldsymbol{\theta}_{\mathcal{D}_N}$ denotes the first-order prediction obtained by a model optimized the first-order objective (e.g., log likelihood) using a dataset $\mathcal{D}_N$ with size $N$, randomly sampled from $P$.

Given this reference distribution, [6, Theorem 3.2] suggests that the optimal second-order distributions derived from EDL methods using "inner loss" are not uniquely defined, [6, Theorem 3.3] states that when there is no regularizer, i.e., $\lambda = 0$, the optimal second-order distribution derived from optimizing the "outer loss" is a Dirac delta function, and [6, Theorem 3.4] demonstrates that for any data generation distribution, there exists a $\lambda > 0$ for which the optimal second-order distribution learned by "outer loss" differs from the reference distribution defined in [6, Definition 3.1].

While [6, Theorem 3.2] falls outside the primary scope of our paper, the arguments from both [6, Theorem 3.3] and [6, Theorem 3.4] can be derived from our Theorem 5.1, which provides an exact characterization of the optimal second-order distribution as follows:

$$p^{\star}(\boldsymbol{\theta}|x) = \frac{p(\boldsymbol{\theta}) \exp(\nu \mathbb{E}_{p(y|x)}[\log p(y|\boldsymbol{\theta})])}{\int p(\boldsymbol{\theta}) \exp(\nu \mathbb{E}_{p(y|x)}[\log p(y|\boldsymbol{\theta})]) \, \mathrm{d}\boldsymbol{\theta}}. \tag{9}$$

For the $\lambda = 0$ scenario described in [6, Theorem 3.3], where $\nu = \lambda^{-1} \to \infty$, the sample $\boldsymbol{\theta}^*$ that maximizes $\mathbb{E}_{p(y|x)}[\log p(y|\boldsymbol{\theta})]$ will exponentially dominate the integral, leading to:

$$p^{\star}(\boldsymbol{\theta}|x) = \frac{p(\boldsymbol{\theta}) \exp(\nu \mathbb{E}_{p(y|x)}[\log p(y|\boldsymbol{\theta})])}{\int p(\boldsymbol{\theta}) \exp(\nu \mathbb{E}_{p(y|x)}[\log p(y|\boldsymbol{\theta})]) \, \mathrm{d}\boldsymbol{\theta}} \xrightarrow{\nu \to \infty} \delta(\boldsymbol{\theta} - \boldsymbol{\theta}^*),$$

where $\boldsymbol{\theta}^* = \arg\max_{\boldsymbol{\theta}} \mathbb{E}_{p(y|x)}[\log p(y|\boldsymbol{\theta})] = \boldsymbol{\eta}(x)$. For the case when $\lambda > 0$ analyzed in [6, Theorem 3.4], our analysis provides the exact optimal second-order distribution in Eq. (9), which clearly demonstrates that it is different from their reference distribution for any specific choice of $\lambda$.

For the classification setting with categorical likelihood $p(y|\boldsymbol{\theta}) = \boldsymbol{\pi}_y$, our analysis in Example 5.2 provides even sharper results. Here, the optimal Dirichlet distribution can be expressed as:

$$p^{\star}(\boldsymbol{\pi}|x) = \mathsf{Dir}(\boldsymbol{\pi}; \boldsymbol{\alpha}_0 + \nu \boldsymbol{\eta}(x))$$

When $\lambda = 0$, $\nu = \lambda^{-1} \to \infty$, the optimal Dirichlet distribution converges to a Dirac function, i.e., $\mathsf{Dir}(\boldsymbol{\pi}; \boldsymbol{\alpha}_0 + \nu \boldsymbol{\eta}(x)) \xrightarrow{\nu \to \infty} \delta(\boldsymbol{\pi} - \boldsymbol{\eta}(x))$. When $\lambda > 0$, the behavior of the learned second-order distribution can be exactly characterized by $\mathsf{Dir}(\boldsymbol{\pi}; \boldsymbol{\alpha}_0 + \lambda^{-1} \boldsymbol{\eta}(x))$.

Hence, both [6, Theorem 3.3] and [6, Theorem 3.4] can be understood as simple corollaries of our Theorem 5.1.

## B  Literature Review for Classical UQ Methods

Classical UQ methods include Bayesian approaches, which model posterior distributions over model parameters and approximate the Bayesian inference in various ways, including variational inference [30, 31], MCMC [32, 33], Monte-Carlo dropout [26], and Laplace approximation [34, 35]. Frequentist methods construct uncertainty sets through techniques such as jackknife [36] and bootstrap [37]. Ensemble methods aggregate decisions of several random models to capture predictive uncertainty [25, 38]. These classical methods suffer from expensive computation cost incurred by either multiple forward passes or training multiple models. More comprehensive review of classical UQ approaches can be found in survey papers [39, 40].

## C  Definition of Uncertainty and Its Measures

In EDL, the uncertainty captured by $p_\psi(\boldsymbol{\pi}|x)$ and $p_\psi(y|x)$ are called the *distributional uncertainty* and *aleatoric (data) uncertainty*, respectively. *distributional uncertainty* is also a kind of epistemic (knowledge) uncertainty that captures the model's lack of knowledge arising from the mismatch in the training distribution and test distribution. That is, the distribution $p_\psi(\boldsymbol{\pi}|x_0)$ should be dispersed if the model is uncertain on the particular query point $x_0$, and sharp otherwise. As the adjective *epistemic* suggests, this uncertainty is supposed to vanish at points in the support of the underlying distribution, when the model has observed infinitely many samples. In contrast, the aleatoric uncertainty captures the inherent uncertainty in $p(y|x)$, and thus must be invariant to the sample size in principle.

There are two standard metrics to measure distributional uncertainty: (1) differential entropy (DEnt) $h_\psi(\boldsymbol{\pi}|x) := -\int p_\psi(\boldsymbol{\pi}|x)\log p_\psi(\boldsymbol{\pi}|x)d\boldsymbol{\pi}$; (2) mutual information (MI) $I_\psi(y;\boldsymbol{\pi}|x) := H(p_\psi(y|x)) - \mathbb{E}_{p_\psi(\boldsymbol{\pi}|x)}[H(p(y|\boldsymbol{\pi}))]$, where $\mathbb{E}_{p_\psi(\boldsymbol{\pi}|x)}[H(p(y|\boldsymbol{\pi}))]$ is usually defined as the *aleatoric (data) uncertainty*. The resulting probabilistic classifier $p_\psi(y|x)$ captures both epistemic and aleatoric uncertainties without distinction, and the *total uncertainty* is often measured via two standard metrics: (1) entropy (Ent) of $p_\psi(y|x)$, i.e., $H(p_\psi(y|x))$; (2) max probability (MaxP) of $p_\psi(y|x)$, i.e., $\max_y p_\psi(y|x)$.

# D  Proof of Theorem 4.1

We prove the four statements in Theorem 4.1 in the following three subsections: the first (PriorNet objectives) is proved in Appendix D.1, second (EDL/MSE objective) and third (VI/ELBO objective) in Appendix D.2, and fourth (PostNet/UCE objective) in Appendix D.3.

## D.1  Prior Network Objective

Prior networks [7, 8] proposed to use the following form of objectives:

$$\mathbb{E}_{p(x,y)}[D(p^{(\nu)}(\boldsymbol{\pi}|y), p_\psi(\boldsymbol{\pi}|x)) + \gamma_{\mathsf{ood}}\mathbb{E}_{p_{\mathsf{ood}}(x)}[D(p(\boldsymbol{\pi}), p_\psi(\boldsymbol{\pi}|x))].$$

Here, $\nu(\in \{10^2, 10^3\})$ and $\gamma_{\mathsf{ood}}$ are hyperparameters.

- The F-KL objective [7] is when $D(p,q) = D(p \parallel q)$. The in-distribution objective can be written as

$$\mathbb{E}_{p(x,y)}[D(p^{(\nu)}(\boldsymbol{\pi}|y) \parallel p_\psi(\boldsymbol{\pi}|x))] = \mathbb{E}_{p(x)}[D(\mathbb{E}_{p(y|x)}[p^{(\nu)}(\boldsymbol{\pi}|y)] \parallel p_\psi(\boldsymbol{\pi}|x))] + (\mathsf{const.}).$$

  This implies that minimizing the F-KL objective function forces the unimodal UQ model $p_\psi(\boldsymbol{\pi}|x)$ to fit the mixture of Dirichlet distributions $\mathbb{E}_{p(y|x)}[p^{(\nu)}(\boldsymbol{\pi}|y)]$. Since the mixture can be multimodal when $p(y|x)$ is not one-hot, or equivalently, if aleatoric uncertainty is nonzero, the F-KL objective will drive the UQ model to spread the probability mass over the simplex, which possibly leads to low accuracy.

- The R-KL objective [8] is when $D(p,q) = D(q \parallel p)$. Unlike the F-KL objective, the in-distribution objective can be written as

$$\mathbb{E}_{p(x,y)}[D(p_\psi(\boldsymbol{\pi}|x) \parallel p^{(\nu)}(\boldsymbol{\pi}|y))] = \mathbb{E}_{p(x)}[D(p_\psi(\boldsymbol{\pi}|x) \parallel \mathsf{Dir}(\boldsymbol{\pi}; \boldsymbol{\alpha}_0 + \nu\boldsymbol{\eta}(x)))] + (\mathsf{const.}).$$

  Since $p_\psi(\boldsymbol{\pi}|x)$ is being fit to another Dirichlet distribution $\mathsf{Dir}(\boldsymbol{\pi}; \boldsymbol{\alpha}_0 + \nu\boldsymbol{\eta}(x))$, it no longer has the issue of the F-KL objective above. We note in passing that Nandy et al. [9] proposed an ad-hoc objective such that $\boldsymbol{\alpha}_\psi(x) < 1$ for OOD $x$'s.

## D.2  Variational Inference and EDL Objective

The variational inference (VI) loss was proposed by Chen et al. [12], Joo et al. [13] in the following form:

$$\ell_{\mathsf{VI}}(\psi; x, y) := \mathbb{E}_{p_\psi(\boldsymbol{\pi}|x)}\left[\log \frac{1}{p(y|\boldsymbol{\pi})}\right] + \lambda D(p_\psi(\boldsymbol{\pi}|x) \parallel p(\boldsymbol{\pi})). \tag{10}$$

Although the original derivation is a bit involved, we can rephrase the key logic in the paper by the following variational relaxation:

$$D(p(x,y) \parallel p(x)p_{\lambda^{-1}}(y)) \leq D(p(x,y)p_\psi(\boldsymbol{\pi}|x) \parallel p(x)p_{\lambda^{-1}}(\boldsymbol{\pi}, y)),$$

where $p^{(\nu)}(y)$ is induced by the tempered distribution $p^{(\nu)}(\boldsymbol{\pi}, y) \propto p(\boldsymbol{\pi})p^\nu(y|\boldsymbol{\pi})$. The inequality is a simple application of a form of data processing inequality [23]. We remark that the original name, "ELBO loss," is a misnomer for this loss function, as the right-hand side simply bounds a constant on the left-hand side, as opposed to the negative ELBO, which bounds the negative log-likelihood.

**Lemma D.1.** *Let $\nu := \lambda^{-1}$. Then, we have*

$$\min_\psi \mathbb{E}_{p(x,y)}[\ell_{\mathsf{VI}}(\psi; x, y)] \equiv \min_\psi \mathbb{E}_{p(x,y)}[D(p_\psi(\boldsymbol{\pi}|x) \parallel p^{(\nu)}(\boldsymbol{\pi}|y))].$$

*Proof.* Note that by the chain rule of KL divergence, we have on the one hand

$$D(p(x,y)p_\psi(\boldsymbol{\pi}|x) \| p(x)p^{(\nu)}(\boldsymbol{\pi},y)) = D(p(x,y) \| p(x)p^{(\nu)}(y)) + \mathbb{E}_{p(x,y)}[D(p_\psi(\boldsymbol{\pi}|x) \| p^{(\nu)}(\boldsymbol{\pi}|y)].$$

Hence,

$$\min_\psi \mathbb{E}_{p(x,y)}[D(p_\psi(\boldsymbol{\pi}|x) \| p^{(\nu)}(\boldsymbol{\pi}|y))] \equiv \min_\psi D(p(x,y)p_\psi(\boldsymbol{\pi}|x) \| p(x)p^{(\nu)}(\boldsymbol{\pi},y)). \qquad (11)$$

On the other hand, we have

$$
\begin{aligned}
&D(p(x,y)p_\psi(\boldsymbol{\pi}|x) \| p(x)p^{(\nu)}(\boldsymbol{\pi},y)) \\
&= \mathbb{E}_{p(x,y)p_\psi(\boldsymbol{\pi}|x)}\left[\log \frac{p(x,y)p_\psi(\boldsymbol{\pi}|x)}{p(x)p^{(\nu)}(\boldsymbol{\pi},y)}\right] \\
&= \mathbb{E}_{p(x,y)p_\psi(\boldsymbol{\pi}|x)}\left[\log \frac{p(x,y)p_\psi(\boldsymbol{\pi}|x)}{p(x)p(\boldsymbol{\pi})p^\nu(y|\boldsymbol{\pi})}\right] + \log B \\
&= \mathbb{E}_{p(x,y)}\left[\nu\mathbb{E}_{p_\psi(\boldsymbol{\pi}|x)}\left[\log \frac{1}{p(y|\boldsymbol{\pi})}\right] + D(p_\psi(\boldsymbol{\pi}|x) \| p(\boldsymbol{\pi}))\right] + D(p(x,y) \| p(x)) + \log B \\
&= \mathbb{E}_{p(x,y)}[\ell_{\mathsf{VI}}(\psi;x,y)] + D(p(x,y) \| p(x)) + \log B.
\end{aligned}
$$

Here, we let $B := \sum_y \int p(\boldsymbol{\pi})p^\nu(y|\boldsymbol{\pi})\,\mathrm{d}\boldsymbol{\pi}$ is the normalization constant, which satisfies $p^{(\nu)}(\boldsymbol{\pi},y) = \frac{1}{B}p(\boldsymbol{\pi})p^\nu(y|\boldsymbol{\pi})$. This implies that

$$\min_\psi D(p(x,y)p_\psi(\boldsymbol{\pi}|x) \| p(x)p^{(\nu)}(\boldsymbol{\pi},y)) \equiv \min_\psi \mathbb{E}_{p(x,y)}[\ell_{\mathsf{VI}}(\psi;x,y)]. \qquad (12)$$

Combining Eq. (12) and Eq. (11), we prove the desired equivalence. $\qquad\square$

### D.3 Uncertainty Cross Entropy Objective

Charpentier et al. [15, 20] proposed to use the uncertainty cross-entropy loss, which is defined as

$$\ell_{\mathsf{UCE}}(\psi;x,y) := \nu\mathbb{E}_{p_\psi(\boldsymbol{\pi}|x)}\left[\log \frac{1}{p(y|\boldsymbol{\pi})}\right] - h(p_\psi(\boldsymbol{\pi}|x)).$$

In [15, 20], this loss function was originally motivated from a general framework for updating belief distributions [22].

As alluded to earlier, however, it is a special instance of the VI objective of Joo et al. [13] when $\boldsymbol{\alpha}_0 = \mathbb{1}$, since $p(\boldsymbol{\pi}) = \mathsf{Dir}(\boldsymbol{\pi};\mathbb{1}) = \frac{1}{B(\mathbb{1})} = (C-1)!$ and thus

$$
\begin{aligned}
D(p_\psi(\boldsymbol{\pi}|x) \| p(\boldsymbol{\pi})) &= \mathbb{E}_{p_\psi(\boldsymbol{\pi}|x)}\left[\log \frac{p_\psi(\boldsymbol{\pi}|x)}{p(\boldsymbol{\pi})}\right] \\
&= -h(p_\psi(\boldsymbol{\pi}|x)) - \mathbb{E}_{p_\psi(\boldsymbol{\pi}|x)}[\log p(\boldsymbol{\pi})] \\
&= -h(p_\psi(\boldsymbol{\pi}|x)) - \log(C-1)!.
\end{aligned}
$$

Since the VI loss Eq. (10) also only uses $\boldsymbol{\alpha}_0 = \mathbb{1}$ in the original paper [13], this implies that the VI loss and the UCE loss are essentially equivalent.

## E  Proof of Theorem 5.1

We prove a stronger result than Theorem 5.1, allowing any observation model for $y$ beyond the categorical case, i.e., $y \in [C]$.

**Theorem E.1.** *For a choice of prior distribution $p(\boldsymbol{\theta})$ and $\lambda > 0$, define*

$$\mathcal{L}(\psi) := \mathbb{E}_{p(x,y)p_\psi(\boldsymbol{\theta}|x)}\left[\log \frac{1}{p(y|\boldsymbol{\theta})}\right] + \lambda\mathbb{E}_{p(x)}[D(p_\psi(\boldsymbol{\theta}|x) \| p(\boldsymbol{\theta}))]. \qquad (13)$$

*Let $\nu := \lambda^{-1}$. Define the tempered likelihood*

$$p^{(\nu)}(\boldsymbol{\theta}|y) := \frac{p^{(\nu)}(\boldsymbol{\theta},y)}{\int p^{(\nu)}(\boldsymbol{\theta},y)\,\mathrm{d}\boldsymbol{\theta}}, \quad \text{where } p^{(\nu)}(\boldsymbol{\theta},y) := \frac{p(\boldsymbol{\theta})p^\nu(y|\boldsymbol{\theta})}{\iint p(\boldsymbol{\theta})p^\nu(y|\boldsymbol{\theta})\,\mathrm{d}\boldsymbol{\theta}\,\mathrm{d}y}.$$

*Then, we have*

$$\min_{\psi} \mathcal{L}(\psi) = \min_{\psi} \mathbb{E}_{p(x,y)}[D(p_\psi(\boldsymbol{\theta}|x) \| p^{(\nu)}(\boldsymbol{\theta}|y))]$$

$$\equiv \min_{\psi} \mathbb{E}_{p(x)}[D(p_\psi(\boldsymbol{\theta}|x) \| p^\star(\boldsymbol{\theta}|x))], \tag{14}$$

*where*

$$p^\star(\boldsymbol{\theta}|x) := \frac{p(\boldsymbol{\theta})\exp(\nu\mathbb{E}_{p(y|x)}[\log p(y|\boldsymbol{\theta})])}{\int p(\boldsymbol{\theta})\exp(\nu\mathbb{E}_{p(y|x)}[\log p(y|\boldsymbol{\theta})])\,\mathrm{d}\boldsymbol{\theta}}.$$

*Proof.* Note that we can rewrite the objective function as

$$\mathcal{L}(\psi) = \lambda\mathbb{E}_{p(x,y)}\Big[D\Big(p_\psi(\boldsymbol{\theta}|x) \,\Big\|\, \frac{p(\boldsymbol{\theta})p^\nu(y|\boldsymbol{\theta})}{\int p(\boldsymbol{\theta})p^\nu(y|\boldsymbol{\theta})\,\mathrm{d}\boldsymbol{\theta}}\Big)\Big] + C$$

$$= \lambda\mathbb{E}_{p(x)}\Big[D\Big(p_\psi(\boldsymbol{\theta}|x) \,\Big\|\, \frac{\exp\Big(\mathbb{E}_{p(y|x)}\Big[\log\frac{p(\boldsymbol{\theta})p^\nu(y|\boldsymbol{\theta})}{\int p(\boldsymbol{\theta})p^\nu(y|\boldsymbol{\theta})\,\mathrm{d}\boldsymbol{\theta}}\Big]\Big)}{\int \exp\Big(\mathbb{E}_{p(y|x)}\Big[\log\frac{p(\boldsymbol{\theta})p^\nu(y|\boldsymbol{\theta})}{\int p(\boldsymbol{\theta})p^\nu(y|\boldsymbol{\theta})\,\mathrm{d}\boldsymbol{\theta}}\Big]\Big)\,\mathrm{d}\boldsymbol{\theta}}\Big)\Big] + C'$$

$$= \lambda\mathbb{E}_{p(x)}\Big[D\Big(p_\psi(\boldsymbol{\theta}|x) \,\Big\|\, \frac{p(\boldsymbol{\theta})\exp(\nu\mathbb{E}_{p(y|x)}[\log p(y|\boldsymbol{\theta})])}{\int p(\boldsymbol{\theta})\exp(\nu\mathbb{E}_{p(y|x)}[\log p(y|\boldsymbol{\theta})])\,\mathrm{d}\boldsymbol{\theta}}\Big)\Big] + C''. \tag{15}$$

Here, $\nu := \frac{1}{\lambda}$ and $C, C'$, and $C''$ are some constants with respect to $\psi$. In particular,

$$C'' = \mathbb{E}_{p(x)}\Big[\log\frac{1}{\int p(\boldsymbol{\theta})\exp(\nu\mathbb{E}_{p(y|x)}[\log p(y|\boldsymbol{\theta})])\,\mathrm{d}\boldsymbol{\theta}}\Big]. \tag{16}$$

This characterizes the optimal $p_\psi(\boldsymbol{\theta}|x)$ under this objective function as

$$p_{\psi^*}(\boldsymbol{\theta}|x) = \frac{p(\boldsymbol{\theta})\exp(\nu\mathbb{E}_{p(y|x)}[\log p(y|\boldsymbol{\theta})])}{\int p(\boldsymbol{\theta})\exp(\nu\mathbb{E}_{p(y|x)}[\log p(y|\boldsymbol{\theta})])\,\mathrm{d}\boldsymbol{\theta}}, \tag{17}$$

which concludes the proof. $\qquad\square$

# F An Extension For General Observation Models

In this section, following the similar exponential family distribution setup, We consider a likelihood model of exponential family distribution over a target variable $y \in \mathbb{R}$ with natural parameters $\boldsymbol{\theta} \in \mathbb{R}^L$

$$p(y|\boldsymbol{\theta}) := h(y)\exp(\boldsymbol{\theta}^\mathsf{T}\mathbf{u}(y) - A(\boldsymbol{\theta})).$$

Here, $h\colon \mathbb{R} \to \mathbb{R}$ is the base measure, $A\colon \mathbb{R}^L \to \mathbb{R}$ is the log-partition function, and $\mathbf{u} \in \mathbb{R} \to \mathbb{R}^L$ is sufficient statistics. Note that the entropy of the parametric distribution is given as $h(p(y|\boldsymbol{\theta})) = A(\boldsymbol{\theta}) - \boldsymbol{\theta}^\mathsf{T}\nabla_{\boldsymbol{\theta}}A(\boldsymbol{\theta}) - \mathbb{E}[\log h(y)]$.

In EDL methods, a distribution over the parameter $\boldsymbol{\theta}$ is assumed to capture uncertainty. A convenient choice is the conjugate prior $p(\boldsymbol{\theta}|\boldsymbol{\xi}, n)$ of the likelihood $p(y|\boldsymbol{\theta})$, which is given as

$$p(\boldsymbol{\theta}|\boldsymbol{\xi}, n) = \eta(\boldsymbol{\xi}, n)\exp(n(\boldsymbol{\theta}^\mathsf{T}\boldsymbol{\xi} - A(\boldsymbol{\theta}))).$$

Here, $\boldsymbol{\xi} \in \mathbb{R}^L$ is the prior parameter, $n \in \mathbb{R}_{>0}$ is the evidence parameter, and $\eta(\boldsymbol{\xi}, n)$ is the normalization constant.

The benefit of the conjugate distribution is in the easy posterior update. Given $N$ observations $y^N$ and prior $p(\boldsymbol{\theta}|\boldsymbol{\xi}_o, n_o)$, the posterior is $p(\boldsymbol{\theta}|\boldsymbol{\xi}_+, n_+)$, where

$$\boldsymbol{\xi}_+ = \frac{n_o\boldsymbol{\xi}_o + \sum_{j=1}^N \mathbf{u}(y_j)}{n_o + N}$$

$$n_+ = n_o + N.$$

Now, we assume that there exists neural networks that, for each point $x$, can approximate the "associated parameters" $n_\psi(x)$ and $\boldsymbol{\xi}_\psi(x)$. We then define the *target* posterior distribution as

$$p_\psi(\boldsymbol{\theta}|x) := p\Big(\boldsymbol{\theta} \,\Big|\, \frac{n_o\boldsymbol{\xi}_o + n_\psi(x)\boldsymbol{\xi}_\psi(x)}{n_o + n_\psi(x)}, n_o + n_\psi(x)\Big).$$

For a choice of prior distribution $p(\boldsymbol{\theta})$ and $\lambda > 0$, define the objective function as

$$\mathcal{L}(\psi) := \mathbb{E}_{p(x,y)p_\psi(\boldsymbol{\theta}|x)}\Big[\log \frac{1}{p(y|\boldsymbol{\theta})}\Big] + \lambda \mathbb{E}_{p(x)}[D(p_\psi(\boldsymbol{\theta}|x) \parallel p(\boldsymbol{\theta}))].$$

This is a generalization of the UCE loss proposed by NatPN [20], as will be detailed below. We note that Theorem 5.1 remains to hold for this general observation model. That is, by minimizing the UCE loss, $p_\psi(\boldsymbol{\theta}|x)$ is encouraged to fit to a fixed target

$$p^\star(\boldsymbol{\theta}|x) = \frac{p(\boldsymbol{\theta})\exp(\nu\mathbb{E}_{p(y|x)}[\log p(y|\boldsymbol{\theta})])}{\int p(\boldsymbol{\theta})\exp(\nu\mathbb{E}_{p(y|x)}[\log p(y|\boldsymbol{\theta})])\,\mathrm{d}\boldsymbol{\theta}}.$$

### F.1 Example: Classification

For classification, we set $\boldsymbol{\theta}(=\boldsymbol{\pi}) \in \Delta^{\mathcal{Y}}$, $p(y|\boldsymbol{\theta}) = \theta_y$ (categorical model), and we have, for $\boldsymbol{\alpha}_o = \mathbb{1}$,

$$p(\boldsymbol{\theta}) = \mathsf{Dir}(\boldsymbol{\theta}; \boldsymbol{\alpha}_o),$$
$$p_\psi(\boldsymbol{\theta}|x) = \mathsf{Dir}(\boldsymbol{\theta}; \boldsymbol{\alpha}_o + n_\psi(x)\boldsymbol{\alpha}_\psi(x)),$$
$$p^\star(\boldsymbol{\theta}|x) = \mathsf{Dir}(\boldsymbol{\theta}; \boldsymbol{\alpha}_0 + \nu\boldsymbol{\eta}(x)) \propto \exp\big(\mathbb{E}_{p(y|x)}\big[\log \mathsf{Dir}(\boldsymbol{\theta}; \boldsymbol{\alpha}_o + \nu\mathbf{e}_y)\big]\big).$$

While the categorical model is typically assumed, a few exceptions exist. Sensoy et al. [41] used $p(y|\boldsymbol{\theta}) = \mathcal{N}(\mathbf{e}_y; \boldsymbol{\theta}, \sigma^2 I)$. Deng et al. [11] used $p(y|\boldsymbol{\theta}, \boldsymbol{\alpha}_\psi(x)) = \mathcal{N}(\mathbf{e}_y; \boldsymbol{\theta}, \sigma^2 \mathcal{I}(\boldsymbol{\alpha}_\psi(x))^{-1})$, where $\mathcal{I}(\boldsymbol{\alpha})$ denotes the Fisher information matrix of the probability model $\mathsf{Dir}(\boldsymbol{\alpha})$, i.e.,

$$\mathcal{I}(\boldsymbol{\alpha}) := \mathbb{E}_{\boldsymbol{\theta}\sim\mathsf{Dir}(\boldsymbol{\alpha})}\Big[\frac{\partial \log \mathsf{Dir}(\boldsymbol{\theta}; \boldsymbol{\alpha})}{\partial\boldsymbol{\alpha}}\frac{\partial \log \mathsf{Dir}(\boldsymbol{\theta}; \boldsymbol{\alpha})}{\partial\boldsymbol{\alpha}}^{\mathsf{T}}\Big].$$

In this modified case, the equivalence in Eq. (14) in Theorem E.1 does not hold. The analysis breaks down since the constant $C''$ in Eq. (15) now becomes dependent on $\psi$, and cannot be ignored in the optimization:

$$C'' = \mathbb{E}_{p(x)}\Big[\log \frac{1}{\mathbb{E}_{p(\boldsymbol{\theta})}[\exp(\nu\mathbb{E}_{p(y|x)}[\log p(y|\boldsymbol{\theta}, \boldsymbol{\alpha}_\psi(x))])]}\Big].$$

Our high-level critique on the EDL methods remains to hold, since the FisherEDL still does not take into account any external uncertainty in the model $\psi$. Therefore, the FisherEDL would fit the meta distribution to some arbitrary target, which could be however potentially better than the simple scaled-and-shifted target. We also note that in our empirical demonstration in Sec. 7, FisherEDL did not show outstanding performance on the downstream task performance.

### F.2 Example: Regression

For regression, we can set $\boldsymbol{\theta} = (\mu, \sigma) \in \mathbb{R} \times \mathbb{R}_{\geq 0}$, $p(y|\boldsymbol{\theta}) = \mathcal{N}(y; \mu, \sigma^2)$. For a prior, normal-inverse-Gamma (NIG) distribution $\mathcal{N}\Gamma^{-1}(\mu, \sigma; \mu_0, \bar{\lambda}, \alpha, \beta)$ is a convenient choice due to its conjugacy to the Gaussian likelihood. Charpentier et al. [20] proposed to use this choice with the UCE loss Eq. (D.3). Malinin et al. [19] considered a multivariate regression with Gaussian likelihood together with the Normal-Wishart distribution, which is a multi-dimensional generalization of the NIG distribution. They used the reverse-KL type objective extending their prior work [8]. In our unified view (Theorem E.1), both objectives are equivalent, and these two approaches would suffer the same issue in its UQ ability.

While the "Deep Evidential Regression" objective proposed by Amini et al. [18] cannot be subsumed by our work, Meinert et al. [42] recently investigated the effectiveness of the EDL method of Amini et al. [18], and also showed that the objective function characterizes a degenerate distribution as its optimal meta distribution, suggesting that its empirical success is likely due to an optimization artifact.

## G Experiment Setup

### G.1 Data Processing

**Details about Synthetic Data.** We create a mixture of Gaussian distribution with mean $[-2, 3]$, $[0, 0]$, and $[2, 3]$, and variance $0.25$. We randomly sample 1000 training data used to train EDL models and use 2000 test data for the generation of visualization plots in Fig. 6 and Fig. 8.

**Data Split of Real Data.** For in-distribution datasets CIFAR10 and CIFAR100, we divide the original training data into two subsets: a training set and a validation set, using an $80\%/20\%$ split ratio. The testing set of these datasets is utilized as in-distribution samples for evaluation. For OOD datasets, we utilize their testing sets as OOD data. To maintain consistency in sample size, we ensured that each OOD dataset contains exactly 10,000 samples, matching the number of in-distribution test samples.

**Details about OOD Data.** For all the OOD datasets, we standardized the dataset images by resizing them to dimensions of 32x32 pixels. Additionally, all gray-scale images were converted to a three-channel format. The following datasets are selected for the OOD detection task:

- Fashion-MNIST: Contains article images of clothes. We use the testing set, which contains 10,000 images.
- SVHN: The Street View House Numbers (SVHN) dataset contains images of house numbers sourced from Google Street View. We use 10,000 images from its testing set as OOD samples.
- TinyImageNet (TIM): As a subset of the larger ImageNet dataset, TIM's validation set, containing 10,000 images, is used as OOD samples.
- Corrupted: This is an artificially created dataset generated by applying perturbations, including Gaussian blurring, pixel permutation, and contrast re-scaling to the original testing images.

### G.2 Model Architecture

**Base Model Architecture.** To ensure a fair comparison, we use the same base model architecture for all EDL methods. We describe the base model architecture for different tasks as follows:

- Synthetic 2-D Gaussian Data: We utilize a simple Multi-Layer Perceptron (MLP) model. This model comprises three linear layers, each with a hidden dimension of $64$, each followed by the ReLU activation function.
- Real Datasets (CIFAR10 and CIFAR100): We adopt the existing model architecture VGG16 [43] and ResNet18 [44] as the base model.

The base model serves as the feature extractor for the EDL model, which extracts the feature with latent dimension 6 for 2-D Gaussian Data and CIFAR10, and extracts the feature with latent dimension 12 for CIFAR100.

**UQ Model Head.** The UQ model of most EDL methods consists of a base model feature extractor and a UQ head. The feature extractor takes the data as input and extracts the latent features. The UQ head then takes the latent feature to parameterize a Dirichlet distribution $\text{Dir}(\boldsymbol{\pi}; \boldsymbol{\alpha}_\psi(x))$. As we discussed in Sec. 4, different EDL method mainly differs from the parametric form of $\boldsymbol{\alpha}_\psi(x)$, either *direct parameterization* or *density parameterization*, which is achieved by using different types of UQ head. We describe the model architecture of these two different UQ heads as follows:

- Direct Parameterization: the UQ head of *direct parameterization* is similar to a typical classification head, which consists of two linear layers with hidden dimension $64$ ($128$ for CIFAR100), equipped with a ReLU activation between the two layers.
- Density Parameterization: *density parameterization* adopts a density model. PostNet [15] uses a radial flow model with a flow length of $6$ to estimate class-wise density, which results in the creation of multiple flow models corresponding to the number of classes in datasets. NatPN [20] uses a single radial flow model with a flow length of $8$ to estimate the density of marginal data distribution $p(x)$.

### G.3 Implementation Details

For all the EDL methods, we use the same base model architecture discussed previously. We elaborate on their common configurations, method-specific configurations, and hyper-parameter as follows.

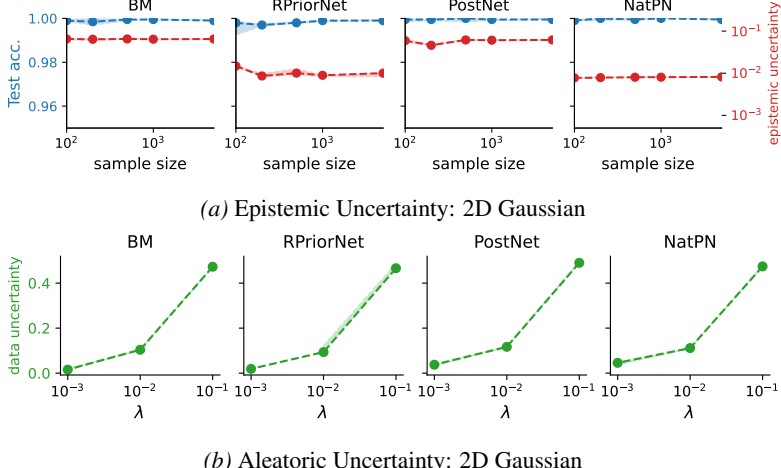

*(a)* Epistemic Uncertainty: 2D Gaussian

*(b)* Aleatoric Uncertainty: 2D Gaussian

*Figure 5:* **Behavior of Uncertainties Learned by EDL methods on Toy Data.** (a) EDL methods learn spurious epistemic uncertainty, wherein uncertainty does not vanish with an increasing number of observed samples, contrary to the fundamental definition of epistemic uncertainty. (b) Instead of a constant, EDL methods learn model-dependent aleatoric uncertainty that depends on hyper-parameter $\lambda$, contrary to the fundamental definition of aleatoric uncertainty.

**Common configurations.** The maximum training epochs are set to $50$, $100$, and $200$ for 2-D Gaussian data, CIFAR10 and CIFAR100, respectively. We train models with the early stopping of patience $10$. The model is evaluated on the validation set with frequency 2, and the optimal model with the lowest validation loss or highest accuracy is returned. The training batch size is set to $64$, $64$, and $256$ for Gaussian data, CIFAR10 and CIFAR100, respectively. We use Adam optimizer without weight decay or learning rate schedule during model optimization. The learning rates of the optimizer are `1e-3,2.5e-4,2.5e-4` for Gaussian data, CIFAR10 and CIFAR100, respectively. The default hyper-parameter $\lambda$ is set to `1e-4` for those EDL methods with regularizer. All reported numbers are averaged over five runs. All experiments are implemented in PyTorch using a Tesla V100 GPU with 32 GB memory.

**Method-specific Configurations.** We describe those EDL methods with additional training configurations and hyper-parameters as follows,

- RPriorNet [8]: This method requires additional OOD data during training. In alignment with the original paper, CIFAR100 serves as the OOD data for CIFAR10, and TinyImageNet is utilized as the OOD data for CIFAR100. We set the OOD regularizer weight $\gamma_{\text{ood}} = 5$ and $\gamma_{\text{ood}} = 1$ for CIFAR10 and CIFAR100, respectively. We realize that both the training loss and validation loss can fluctuate significantly due to the conflict between in-distribution and OOD data optimization. In this case, we use validation accuracy as the criterion to save the optimal model, making sure the model is well optimized.

- NatPN [20]: NatPN uses a single density model with flow length 8 and latent feature dimension 16 to parameterize density. The method also requires "warm-up training" and "fine-tuning" of the density model, where we set warm-up epochs to be 3 and fine-tuning epochs to 100, as the paper's official implementation suggests.

- Fisher-EDL [11]: This method requires two regularizers, one is the standard r-KL regularizer, the other their proposed "Fisher Information" regularizer in the training objective. As the paper suggest, we set the corresponding hyper-parameter $\lambda_2 = $ `5e-2` and $\lambda_2 = $ `5e-3` for CIFAR10 and CIFAR100, respectively, and set $\lambda_1 = 1$ for both datasets.

- END2 [16]: To obtain the ensemble samples, we train $100$ ensemble models using standard cross entropy loss. In alignment with the original paper, we also use the temperature annealing technique, which is the key technique to ensure the distillation training works smoothly. The temperature will be initialized as a large value and then gradually decay to 1 after certain epochs. We set the initial temperature to 5 for both datasets and set the decay epochs to 30 and 60 for CIFAR10 and CIFAR100, respectively.

- S2D [17]: We train a single model with random dropout and then run inference of the model to obtain 100 samples. We also use the temperature annealing technique with the same configurations as END2 for the distillation training.

- Bootstrap Distillation (Ours): First, we randomly sample 100 subsets of training data from the original training set with an $80\%/20\%$ split ratio. Then, for each dataset, we train a bootstrap model using standard cross entropy loss, resulting 100 bootstrap models. For the distillation stage, we also apply the temperature annealing technique with the same configurations as END2.

# H    Additional Experiment Results

## H.1    Omitted Results in Sec. 5.1

The behavior of both epistemic uncertainty and aleatoric uncertainty learned by EDL methods on 2D Gaussian data is shown in Fig. 5. We observe similar behavior as real data. EDL methods cannot quantify either epistemic or aleatoric uncertainty in a faithful way.

## H.2    Ablation Study in Sec. 5.2

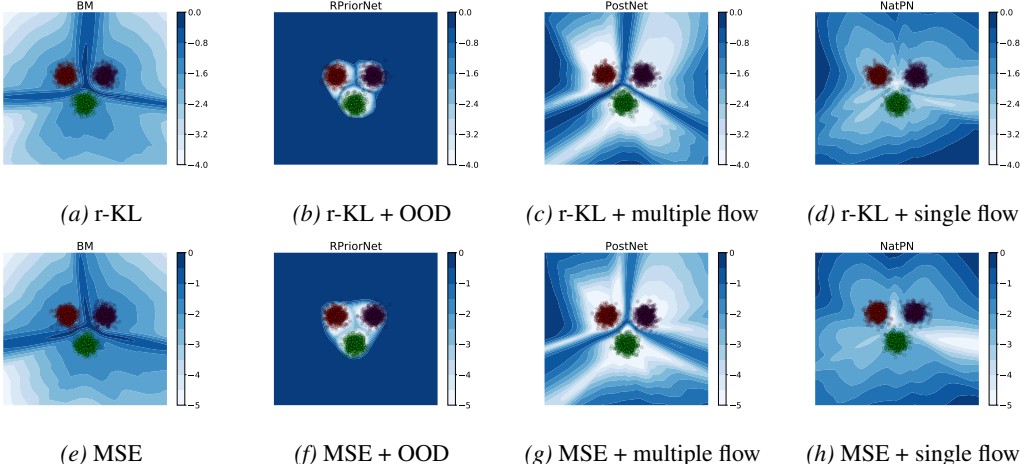

*(a)* r-KL       *(b)* r-KL + OOD       *(c)* r-KL + multiple flow       *(d)* r-KL + single flow

*(e)* MSE       *(f)* MSE + OOD       *(g)* MSE + multiple flow       *(h)* MSE + single flow

*Figure 6:* **Visualization of Epistemic Uncertainty on 2D Gaussian Data.** The first row $(a)$-$(d)$ corresponds to differential entropy quantified by EDL models trained using their original proposed training objective (reverse-KL), and the second row $(e)$-$(h)$ corresponds to epistemic uncertainty quantified by same EDL methods ablated with MSE objective in Eq. (18). This ablation study implies that the specific training objectives has no actual impact on the methods' ability to quantify uncertainty. Other techniques, such as using density estimation (PostNet $(c, g)$, NatPN $(d, h)$), and using OOD data during training (RPriorNet $(b, f)$), play a more significant role in EDL methods' UQ performance. Without auxiliary techniques, BM $(a, e)$ cannot distinguish in-distribution and OOD regions. Similar behavior holds for total uncertainty quantification (see Fig. 8).

There could be many other Dirichlet-framework-independent objectives that can capture similar behavior. For instance, we can consider a simple variant of MSE loss as follows:

$$\min_{\psi} \mathbb{E}_{p(x,y)}[\| \log(\boldsymbol{\alpha}_{\psi}(x)) - \log(\boldsymbol{\alpha}_0 + \lambda^{-1}\mathbf{e}_y)\|^2].  \tag{18}$$

The global minimum of optimizing this heuristic loss function can be characterized by $\log(\boldsymbol{\alpha}_{\psi^\star}(x)) = \mathbb{E}_{p(y|x)}[\log(\boldsymbol{\alpha}_0 + \lambda^{-1}\mathbf{e}_y)]$, which implies $\boldsymbol{\alpha}_{\psi}(x) \approx \boldsymbol{\alpha}_0 + \lambda^{-1}\boldsymbol{\eta}(x)$ under the assumption of a sharp true label distribution $p(y|x)$. Independent of the training procedure, we can artificially define $-\log \mathbb{1}_C^\mathsf{T} \boldsymbol{\alpha}_{\psi}(x)$ as an epistemic uncertainty metric, while the entropy of normalized model output $H(\boldsymbol{\alpha}_{\psi}(x)/\mathbb{1}_C^\mathsf{T} \boldsymbol{\alpha}_{\psi}(x))$ can represent total uncertainty.

We conduct an ablation study by replacing the reverse-KL objective with the MSE objective in Eq. (18) for model training. First, we provide visualization of epistemic uncertainty, aleatoric uncertainty, and total uncertainty on 2D Gaussian data quantified by different EDL methods in Fig. 6, Fig. 7,

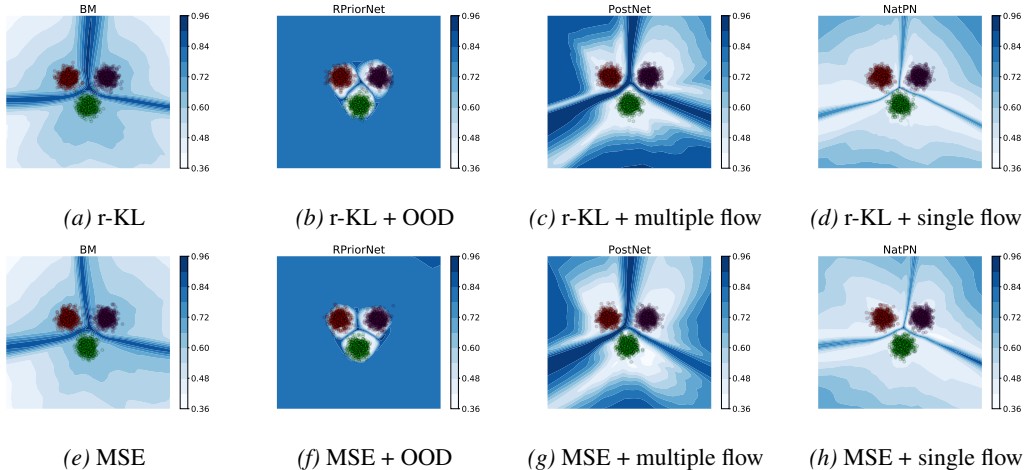

*(a)* r-KL     *(b)* r-KL + OOD     *(c)* r-KL + multiple flow     *(d)* r-KL + single flow

*(e)* MSE     *(f)* MSE + OOD     *(g)* MSE + multiple flow     *(h)* MSE + single flow

*Figure 7:* **Visualization of Aleatoric Uncertainty on 2D Gaussian Data.** The first row $(a - d)$ corresponds to the aleatoric uncertainty quantified by EDL models trained using their original proposed training objective (reverse-KL), and the second row $(e - h)$ corresponds to aleatoric uncertainty quantified by the same EDL methods ablated with MSE objective in the equation Eq. (18). Based on these results, We can draw a similar conclusion as Fig. 6: specific training objective has no actual impact on the methods' ability to quantify aleatoric uncertainty either.

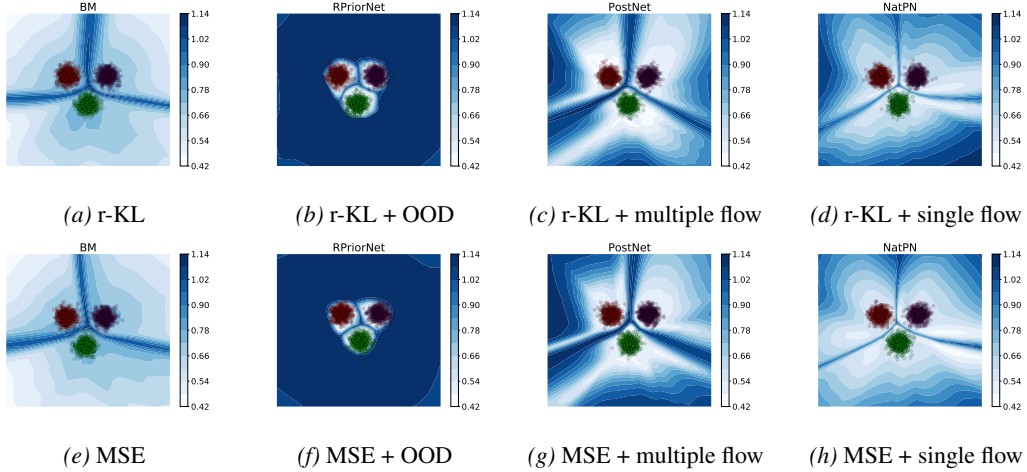

*(a)* r-KL     *(b)* r-KL + OOD     *(c)* r-KL + multiple flow     *(d)* r-KL + single flow

*(e)* MSE     *(f)* MSE + OOD     *(g)* MSE + multiple flow     *(h)* MSE + single flow

*Figure 8:* **Visualization of Total Uncertainty on 2D Gaussian Data.** Total uncertainty is measured using entropy. We can draw a similar conclusion as Fig. 6 and Fig. 7.

and Fig. 8, respectively. Next, we extend the ablation study to OOD detection (using epistemic uncertainty) and ambiguous data detection (using aleatoric uncertainty) tasks using real data in Fig. 9 and Fig. 10, respectively. The results indicate that EDL methods exhibit similar behavior by using reverse KL or MSE objectives.

### H.3   Ablation Study in Sec. 5.3

**Empirical Finding 1. OOD-Data-Dependent Methods are Sensitive to Model Architectures**

Firstly, we remark on the conflicting empirical findings regarding the Prior Network [7, 8] as reported in current EDL literature. Baseline results of [8] on the OOD detection task, reproduced by [15, 11], deviate a lot from the original paper's results. Upon closer investigation, we can attribute these conflicting results to nuances in different model architectures, i.e., the use of BatchNorm has a significant impact when incorporating OOD data during training. Specifically, we conduct an ablation study on the model architectures of RPriorNet, using VGG-16 and ResNet as backbone models, and

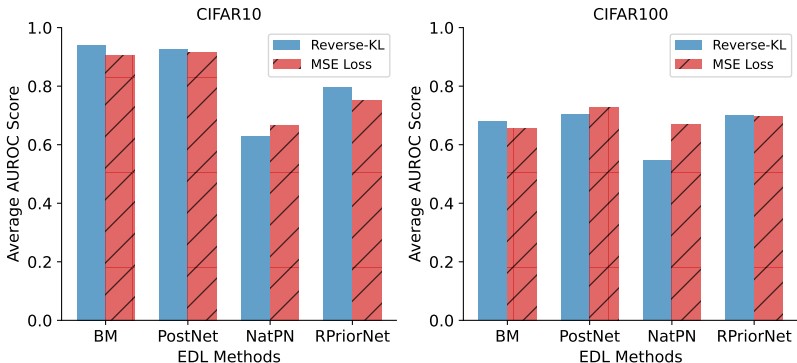

*Figure 9:* **Comparison of OOD Detection Performance of EDL methods with Original Objective v.s. MSE Objective.** The EDL models trained using MSE objective in Eq. (18) demonstrate comparable performance. These results further justify that the specific training objective has no actual impact on the downstream task performance.

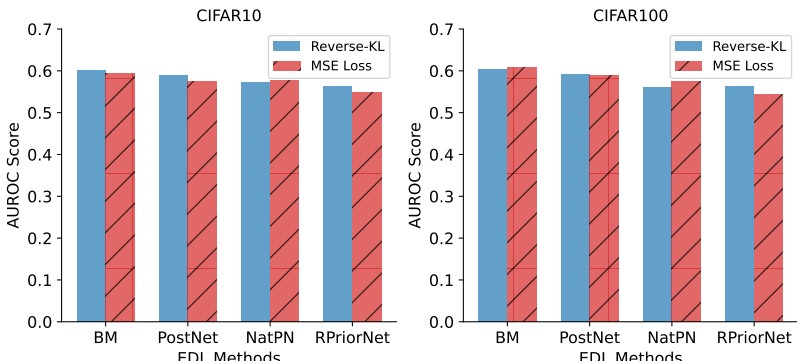

*Figure 10:* **Comparison of Ambiguous Data Detection Performance of EDL methods with Original Objective v.s. MSE Objective.** We can draw a similar conclusion as Fig. 9.

compare their performance with and without BatchNorm layers removed. As illustrated in Figures 11, the OOD detection performance of RPriorNet shows sensitivity to the model architecture, with the removal of BatchNorm layers leading to performance improvement. This observation further reflects the limitations of utilizing OOD data during training, particularly for large-scale models in which BatchNorm is usually employed.

**Empirical Finding 2. Density Models May Not Perform Well with Moderate Dimensionality**

Charpentier et al. [15, 20] claim that utilizing density parametrization can effectively address the OOD detection task without the need to observe OOD data during training. This seems to be intuitive, since a perfect density estimator should be capable of identifying OOD data, as OOD data are typically from low-density regions. However, density estimation itself may encounter challenges, particularly in high-dimensional spaces. To validate this conjecture, we train PostNet [15] with a latent feature dimension of 2. As illustrated in Fig. 12, we visualize the epistemic uncertainty (measured by differential entropy) in the latent feature space and plot the features of both in-distribution and OOD data. Despite the density model effectively generating higher uncertainty for OOD regions, the OOD data are mapped to the same region as the in-distribution data. This phenomenon, known as model collapse [45], limits the model's capability to distinguish between in-distribution and OOD data.

**H.4 Bootstrap Distillation Method Faithfully quantify Epistemic Uncertainty**

The comparison of epistemic uncertainty quantified by existing EDL methods and our proposed Bootstrap Distillation method is shown in Fig. 13. Compared to existing EDL methods, our proposed

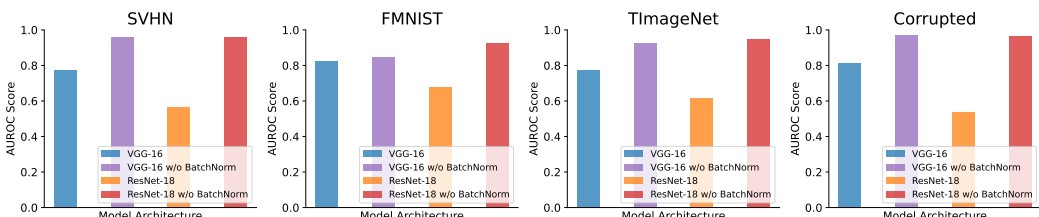

*Figure 11:* **OOD Detection Performance of RPriorNet with Different Model Architecture**. RPriorNet is trained using CIFAR10 as in-distribution data and CIFAR100 as OOD data. The OOD detection performance (showing four OOD datasets) is sensitive to the choice of model architecture.

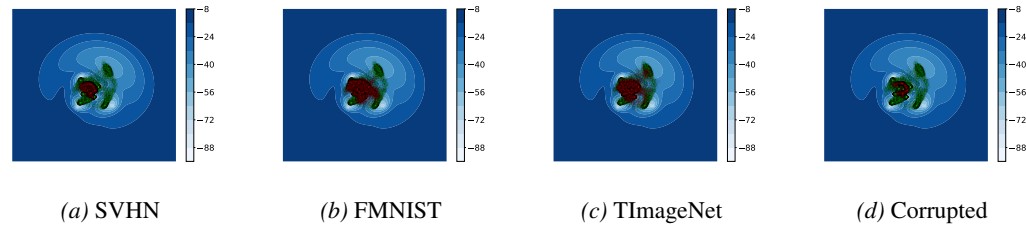

*(a)* SVHN      *(b)* FMNIST      *(c)* TImageNet      *(d)* Corrupted

*Figure 12:* **Visualization of Epistemic Uncertainty in PostNet's 2D Latent Feature Space.** PostNet leverages a flow-based density estimation model, which outputs low (high) uncertainty for in-distribution (OOD) regions as expected. However, given that the input data is high-dimensional, PostNet suffers from the feature collapse issue that mapping **OOD data** to the same region as **ID data** in the latent space, making them indistinguishable.

Bootstrap Distillation method demonstrates monotonically decreasing and eventually vanishing uncertainty, consistent with the dictionary definition of epistemic uncertainty.

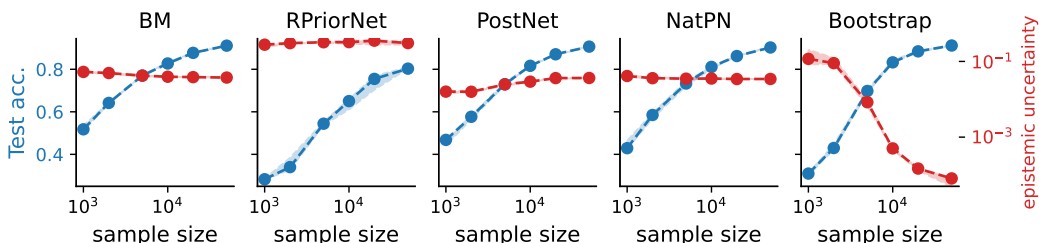

*Figure 13:* **Epistemic Uncertainty and Test Accuracy v.s. Number of Training Data.** Compared to other EDL methods, our proposed Bootstrap Distillation method can faithfully quantify epistemic uncertainty, i.e., the uncertainty is monotonically decreasing with increasing number of data.

## H.5    Bootstrap Distillation Method Benefits from More Samples

As discussed in Sec. 6, the Bootstrap Distillation method needs the sampling of multiple bootstrap samples, with each requiring the training of a model using randomly sampled datasets. To understand the impact of varying the number of samples 5, 10, 20, 50, 100, we conduct an empirical investigation on the OOD detection performance of Bootstrap Distillation. The results shown in Fig. 14 reveal that Bootstrap Distillation indeed benefits from a larger number of samples, particularly in more complex OOD detection tasks using CIFAR100 as the in-distribution data.

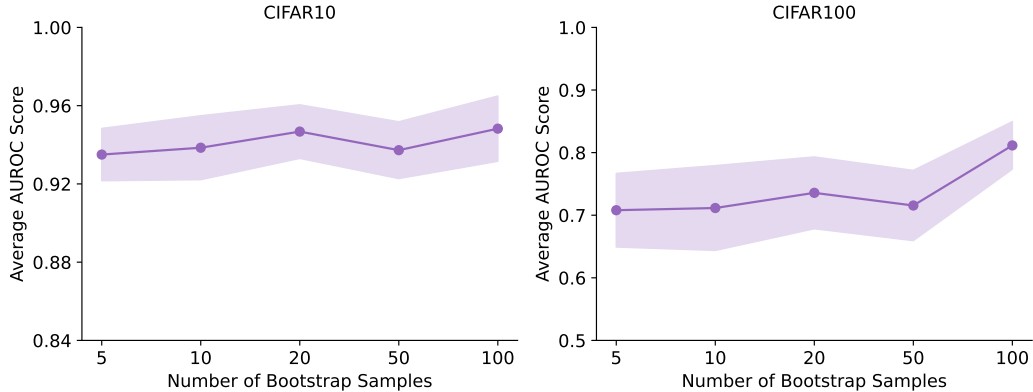

*Figure 14:* **Bootstrap Distillation's OOD Detection Performance v.s. Number of Bootstrap Samples.** The $x$-axis represents the number of bootstrap samples, and the y-axis represents the Average AUROC score of OOD detection tasks. Bootstrap Distillation's UQ performance will improve by obtaining more bootstrap samples, but more computational cost is the price needs to pay.

*Table 2:* **OOD detection AUROC score.** MI and Dent stand for Mutual Information and Differential Entropy.

| ID Data | Method | Metric | SVHN | FMNIST | TImageNet | Corrupted |
|---|---|---|---|---|---|---|
| **CIFAR10** | RPriorNet | MI | $72.6_{\pm 6.9}$ | $79.8_{\pm 2.9}$ | $74.0_{\pm 1.9}$ | $77.6_{\pm 9.8}$ |
| | | Dent | $77.6_{\pm 5.8}$ | $82.7_{\pm 2.1}$ | $77.4_{\pm 1.5}$ | $81.3_{\pm 7.1}$ |
| | BM | MI | $97.5_{\pm 0.6}$ | $93.6_{\pm 0.6}$ | $87.4_{\pm 1.2}$ | $98.8_{\pm 0.4}$ |
| | | Dent | $97.0_{\pm 0.8}$ | $93.3_{\pm 0.5}$ | $87.7_{\pm 1.1}$ | $98.3_{\pm 0.4}$ |
| | PostNet | MI | $94.4_{\pm 1.0}$ | $89.8_{\pm 1.6}$ | $86.3_{\pm 0.3}$ | $96.8_{\pm 0.8}$ |
| | | Dent | $95.5_{\pm 1.0}$ | $90.6_{\pm 1.2}$ | $86.8_{\pm 0.3}$ | $97.8_{\pm 0.6}$ |
| | NatPN | MI | $17.8_{\pm 2.9}$ | $21.0_{\pm 2.7}$ | $26.3_{\pm 3.2}$ | $14.9_{\pm 2.1}$ |
| | | Dent | $67.0_{\pm 6.7}$ | $60.6_{\pm 4.1}$ | $54.6_{\pm 2.6}$ | $69.7_{\pm 5.8}$ |
| | EDL | MI | $96.2_{\pm 1.1}$ | $90.3_{\pm 0.6}$ | $86.3_{\pm 0.4}$ | $98.7_{\pm 0.8}$ |
| | | Dent | $96.5_{\pm 0.9}$ | $90.3_{\pm 0.6}$ | $86.9_{\pm 0.4}$ | $98.8_{\pm 0.3}$ |
| | Fisher-EDL | MI | $91.3_{\pm 0.3}$ | $86.9_{\pm 0.8}$ | $86.1_{\pm 0.6}$ | $95.1_{\pm 1.0}$ |
| | | Dent | $93.7_{\pm 1.3}$ | $88.1_{\pm 1.1}$ | $86.6_{\pm 0.8}$ | $97.1_{\pm 0.7}$ |
| | END2 | MI | $95.8_{\pm 0.4}$ | $92.7_{\pm 0.7}$ | $89.0_{\pm 0.4}$ | $95.7_{\pm 0.5}$ |
| | | Dent | $96.9_{\pm 0.2}$ | $93.4_{\pm 0.7}$ | $87.1_{\pm 0.8}$ | $96.9_{\pm 0.4}$ |
| | S2D | MI | $89.1_{\pm 2.7}$ | $87.9_{\pm 0.6}$ | $83.9_{\pm 0.6}$ | $90.3_{\pm 3.5}$ |
| | | Dent | $93.3_{\pm 1.5}$ | $90.0_{\pm 0.4}$ | $87.0_{\pm 0.4}$ | $93.9_{\pm 0.2}$ |
| | **Bootstrap Distill** | MI | $97.2_{\pm 0.1}$ | $94.4_{\pm 0.5}$ | $89.6_{\pm 0.3}$ | $98.2_{\pm 0.2}$ |
| | | Dent | $97.8_{\pm 0.1}$ | $94.3_{\pm 0.6}$ | $87.6_{\pm 0.4}$ | $98.4_{\pm 0.2}$ |
| **ID Data** | **Method** | **Metric** | **SVHN** | **FMNIST** | **TImageNet** | **Corrupted** |
| **CIFAR100** | RPriorNet | MI | $74.8_{\pm 1.8}$ | $73.1_{\pm 4.6}$ | $61.6_{\pm 2.5}$ | $72.9_{\pm 5.3}$ |
| | | Dent | $74.3_{\pm 1.6}$ | $73.7_{\pm 4.5}$ | $63.1_{\pm 2.2}$ | $68.9_{\pm 6.4}$ |
| | BM | MI | $71.5_{\pm 9.6}$ | $74.6_{\pm 6.8}$ | $71.4_{\pm 5.5}$ | $64.8_{\pm 6.9}$ |
| | | Dent | $68.5_{\pm 10.2}$ | $74.2_{\pm 6.7}$ | $70.2_{\pm 6.0}$ | $58.6_{\pm 6.8}$ |
| | PostNet | MI | $75.9_{\pm 5.6}$ | $70.8_{\pm 2.4}$ | $64.4_{\pm 1.6}$ | $57.9_{\pm 11.6}$ |
| | | Dent | $79.4_{\pm 5.0}$ | $74.7_{\pm 1.8}$ | $72.0_{\pm 0.9}$ | $55.5_{\pm 10.6}$ |
| | NatPN | MI | $56.4_{\pm 5.6}$ | $41.6_{\pm 4.1}$ | $43.5_{\pm 1.5}$ | $53.2_{\pm 5.9}$ |
| | | Dent | $64.7_{\pm 4.4}$ | $47.0_{\pm 4.5}$ | $49.6_{\pm 0.9}$ | $57.0_{\pm 3.5}$ |
| | EDL | MI | $68.7_{\pm 0.4}$ | $74.8_{\pm 2.6}$ | $73.7_{\pm 0.5}$ | $50.5_{\pm 6.5}$ |
| | | Dent | $68.2_{\pm 3.4}$ | $74.4_{\pm 2.5}$ | $73.5_{\pm 0.5}$ | $49.9_{\pm 6.3}$ |
| | Fisher-EDL | MI | $71.3_{\pm 2.6}$ | $71.0_{\pm 3.2}$ | $74.8_{\pm 0.6}$ | $54.8_{\pm 7.5}$ |
| | | Dent | $71.5_{\pm 2.9}$ | $69.6_{\pm 3.3}$ | $75.4_{\pm 0.3}$ | $53.6_{\pm 6.3}$ |
| | END2 | MI | $78.1_{\pm 2.9}$ | $79.6_{\pm 1.5}$ | $76.6_{\pm 0.4}$ | $68.4_{\pm 2.6}$ |
| | | Dent | $82.0_{\pm 2.4}$ | $85.0_{\pm 0.6}$ | $83.2_{\pm 0.2}$ | $73.4_{\pm 1.6}$ |
| | S2D | MI | $62.1_{\pm 2.2}$ | $69.8_{\pm 0.9}$ | $71.5_{\pm 0.2}$ | $31.6_{\pm 1.1}$ |
| | | Dent | $50.5_{\pm 0.2}$ | $50.4_{\pm 0.3}$ | $50.9_{\pm 0.3}$ | $50.3_{\pm 0.3}$ |
| | **Bootstrap Distill** | MI | $77.9_{\pm 1.3}$ | $83.3_{\pm 1.6}$ | $78.2_{\pm 0.5}$ | $53.4_{\pm 3.0}$ |
| | | Dent | $85.6_{\pm 1.0}$ | $86.8_{\pm 1.2}$ | $84.1_{\pm 0.2}$ | $68.1_{\pm 3.1}$ |

## H.6 Detailed Results on UQ Downstream Tasks

The complete AUROC and AUPR results of OOD data detection are shown in Table 2 and Table 3, respectively. The complete results of selective classification are shown in Table 4.

Table 3: **OOD detection AUPR score.** MI and Dent stand for Mutual Information and Differential Entropy.

| ID Data | Method | Metric | SVHN | FMNIST | TImageNet | Corrupted |
|---|---|---|---|---|---|---|
| **CIFAR10** | RPriorNet | MI | $68.2_{\pm7.5}$ | $75.6_{\pm3.0}$ | $69.3_{\pm2.9}$ | $73.6_{\pm9.6}$ |
| | | Dent | $73.5_{\pm6.3}$ | $79.9_{\pm1.9}$ | $72.9_{\pm1.9}$ | $75.4_{\pm8.9}$ |
| | BM | MI | $96.9_{\pm0.8}$ | $92.3_{\pm0.4}$ | $85.3_{\pm1.3}$ | $98.2_{\pm0.6}$ |
| | | Dent | $95.0_{\pm1.7}$ | $90.5_{\pm1.2}$ | $84.2_{\pm1.6}$ | $96.4_{\pm1.2}$ |
| | PostNet | MI | $90.9_{\pm1.6}$ | $86.6_{\pm2.3}$ | $82.3_{\pm0.6}$ | $93.3_{\pm2.2}$ |
| | | Dent | $93.5_{\pm1.2}$ | $88.5_{\pm1.3}$ | $83.7_{\pm0.4}$ | $96.1_{\pm1.2}$ |
| | NatPN | MI | $43.9_{\pm2.9}$ | $40.4_{\pm2.0}$ | $38.7_{\pm1.2}$ | $44.9_{\pm2.6}$ |
| | | Dent | $70.2_{\pm5.4}$ | $65.1_{\pm3.2}$ | $59.2_{\pm1.9}$ | $73.6_{\pm3.9}$ |
| | EDL | MI | $93.7_{\pm1.9}$ | $87.6_{\pm0.9}$ | $82.6_{\pm0.5}$ | $97.6_{\pm0.6}$ |
| | | Dent | $94.4_{\pm1.6}$ | $87.7_{\pm0.8}$ | $83.4_{\pm0.4}$ | $98.0_{\pm0.4}$ |
| | Fisher-EDL | MI | $82.2_{\pm1.3}$ | $80.5_{\pm1.1}$ | $80.9_{\pm0.6}$ | $89.0_{\pm2.7}$ |
| | | Dent | $88.2_{\pm2.7}$ | $84.7_{\pm1.5}$ | $82.5_{\pm1.1}$ | $94.0_{\pm1.7}$ |
| | END2 | MI | $93.3_{\pm0.2}$ | $90.9_{\pm0.8}$ | $84.3_{\pm0.7}$ | $91.7_{\pm1.0}$ |
| | | Dent | $95.1_{\pm0.2}$ | $91.8_{\pm0.7}$ | $83.7_{\pm0.8}$ | $93.8_{\pm0.9}$ |
| | S2D | MI | $83.0_{\pm3.9}$ | $82.6_{\pm0.9}$ | $77.7_{\pm0.7}$ | $83.0_{\pm5.2}$ |
| | | Dent | $89.0_{\pm2.4}$ | $86.4_{\pm0.7}$ | $82.4_{\pm0.4}$ | $87.5_{\pm3.6}$ |
| | **Bootstrap Distill** | MI | $94.6_{\pm0.4}$ | $92.5_{\pm0.7}$ | $85.6_{\pm0.5}$ | $95.2_{\pm0.5}$ |
| | | Dent | $95.7_{\pm0.3}$ | $92.5_{\pm0.8}$ | $84.5_{\pm0.6}$ | $96.0_{\pm0.4}$ |

| ID Data | Method | Metric | SVHN | FMNIST | TImageNet | Corrupted |
|---|---|---|---|---|---|---|
| **CIFAR100** | RPriorNet | MI | $70.3_{\pm1.3}$ | $69.6_{\pm4.9}$ | $58.7_{\pm2.1}$ | $71.6_{\pm6.2}$ |
| | | Dent | $69.1_{\pm1.4}$ | $69.2_{\pm4.6}$ | $59.2_{\pm1.6}$ | $68.5_{\pm6.7}$ |
| | BM | MI | $67.3_{\pm7.8}$ | $68.9_{\pm5.8}$ | $67.2_{\pm5.5}$ | $59.3_{\pm5.5}$ |
| | | Dent | $65.0_{\pm7.5}$ | $68.1_{\pm5.1}$ | $66.4_{\pm5.5}$ | $53.7_{\pm4.3}$ |
| | PostNet | MI | $69.7_{\pm6.6}$ | $64.4_{\pm2.6}$ | $57.5_{\pm1.4}$ | $59.7_{\pm10.3}$ |
| | | Dent | $75.4_{\pm5.5}$ | $69.3_{\pm2.3}$ | $66.2_{\pm1.2}$ | $57.6_{\pm9.5}$ |
| | NatPN | MI | $54.3_{\pm5.2}$ | $44.1_{\pm2.5}$ | $45.5_{\pm1.1}$ | $50.2_{\pm3.7}$ |
| | | Dent | $59.8_{\pm4.7}$ | $47.0_{\pm2.7}$ | $48.9_{\pm0.8}$ | $52.0_{\pm2.3}$ |
| | EDL | MI | $63.9_{\pm3.8}$ | $69.7_{\pm2.7}$ | $68.9_{\pm0.5}$ | $48.5_{\pm3.8}$ |
| | | Dent | $63.1_{\pm3.3}$ | $68.8_{\pm2.6}$ | $68.6_{\pm0.5}$ | $47.8_{\pm3.6}$ |
| | Fisher-EDL | MI | $64.0_{\pm3.1}$ | $66.6_{\pm3.1}$ | $69.6_{\pm1.3}$ | $53.0_{\pm6.1}$ |
| | | Dent | $65.0_{\pm3.2}$ | $64.9_{\pm3.2}$ | $71.0_{\pm0.7}$ | $49.9_{\pm4.1}$ |
| | END2 | MI | $71.1_{\pm3.6}$ | $70.3_{\pm1.3}$ | $70.5_{\pm0.4}$ | $57.9_{\pm2.3}$ |
| | | Dent | $75.1_{\pm2.7}$ | $77.0_{\pm0.4}$ | $78.2_{\pm0.3}$ | $63.5_{\pm2.4}$ |
| | S2D | MI | $58.5_{\pm2.0}$ | $63.9_{\pm0.7}$ | $68.6_{\pm0.3}$ | $38.0_{\pm0.4}$ |
| | | Dent | $46.4_{\pm0.4}$ | $45.6_{\pm0.1}$ | $45.8_{\pm0.1}$ | $52.5_{\pm0.7}$ |
| | **Bootstrap Distill** | MI | $68.5_{\pm1.6}$ | $73.6_{\pm2.5}$ | $71.9_{\pm0.8}$ | $46.6_{\pm1.4}$ |
| | | Dent | $77.7_{\pm1.0}$ | $78.6_{\pm2.0}$ | $79.4_{\pm0.4}$ | $55.8_{\pm2.3}$ |

Table 4: **Selective classification Test Accuracy, AUROC, and AUPR score.** Ent and MaxP stand for Entropy and Max Probability, respectively.

| Method | Metric | CIFAR10 | | | CIFAR100 | | |
|---|---|---|---|---|---|---|---|
| | | Test Acc | AUROC | AUPR | Test Acc | AUROC | AUPR |
| RPriorNet | Ent | $85.4_{\pm1.4}$ | $86.0_{\pm0.4}$ | $49.7_{\pm2.0}$ | $57.9_{\pm1.3}$ | $80.8_{\pm0.5}$ | $72.2_{\pm0.9}$ |
| | MaxP | $85.4_{\pm1.4}$ | $86.6_{\pm0.3}$ | $51.3_{\pm1.6}$ | $57.9_{\pm1.3}$ | $81.8_{\pm0.3}$ | $74.0_{\pm0.6}$ |
| BM | Ent | $88.7_{\pm0.1}$ | $90.1_{\pm0.2}$ | $51.5_{\pm0.7}$ | $54.3_{\pm1.2}$ | $81.8_{\pm0.1}$ | $76.0_{\pm0.7}$ |
| | MaxP | $88.7_{\pm0.1}$ | $90.0_{\pm0.2}$ | $51.6_{\pm0.7}$ | $54.3_{\pm1.2}$ | $82.6_{\pm0.1}$ | $76.9_{\pm0.7}$ |
| PostNet | Ent | $88.4_{\pm0.1}$ | $89.2_{\pm0.2}$ | $50.4_{\pm0.6}$ | $58.0_{\pm0.8}$ | $81.6_{\pm0.2}$ | $74.1_{\pm1.0}$ |
| | MaxP | $88.4_{\pm0.1}$ | $89.2_{\pm0.2}$ | $50.7_{\pm0.8}$ | $58.0_{\pm0.8}$ | $82.6_{\pm0.2}$ | $75.4_{\pm0.5}$ |
| NatPN | Ent | $83.2_{\pm0.6}$ | $86.2_{\pm0.2}$ | $53.5_{\pm0.5}$ | $56.0_{\pm0.4}$ | $80.8_{\pm0.2}$ | $73.8_{\pm0.5}$ |
| | MaxP | $83.2_{\pm0.6}$ | $86.4_{\pm0.2}$ | $54.4_{\pm0.6}$ | $56.0_{\pm0.4}$ | $82.2_{\pm0.2}$ | $76.2_{\pm0.3}$ |
| EDL | Ent | $89.9_{\pm0.1}$ | $89.7_{\pm0.2}$ | $49.2_{\pm0.7}$ | $62.9_{\pm0.3}$ | $84.5_{\pm0.2}$ | $74.1_{\pm0.3}$ |
| | MaxP | $89.9_{\pm0.1}$ | $89.7_{\pm0.2}$ | $48.8_{\pm0.7}$ | $62.9_{\pm0.3}$ | $84.9_{\pm0.2}$ | $74.9_{\pm0.3}$ |
| Fisher-EDL | Ent | $88.2_{\pm0.4}$ | $90.4_{\pm0.1}$ | $53.7_{\pm0.9}$ | $55.9_{\pm0.8}$ | $85.8_{\pm0.2}$ | $79.6_{\pm0.3}$ |
| | MaxP | $88.2_{\pm0.4}$ | $90.4_{\pm0.1}$ | $53.8_{\pm0.8}$ | $55.9_{\pm0.8}$ | $86.1_{\pm0.2}$ | $80.1_{\pm0.3}$ |
| END2 | Ent | $89.4_{\pm0.1}$ | $89.8_{\pm0.3}$ | $50.6_{\pm1.0}$ | $65.0_{\pm0.2}$ | $83.0_{\pm0.1}$ | $69.3_{\pm0.2}$ |
| | MaxP | $89.4_{\pm0.1}$ | $89.6_{\pm0.3}$ | $51.1_{\pm1.0}$ | $65.0_{\pm0.2}$ | $85.3_{\pm0.1}$ | $72.6_{\pm0.2}$ |
| S2D | Ent | $89.1_{\pm0.1}$ | $89.4_{\pm0.1}$ | $50.5_{\pm0.2}$ | $61.0_{\pm0.1}$ | $83.4_{\pm0.1}$ | $72.7_{\pm0.2}$ |
| | MaxP | $89.1_{\pm0.1}$ | $89.3_{\pm0.1}$ | $51.0_{\pm0.5}$ | $61.0_{\pm0.1}$ | $84.4_{\pm0.1}$ | $74.2_{\pm0.2}$ |
| **Bootstrap Distill** | Ent | $89.4_{\pm0.1}$ | $90.3_{\pm0.2}$ | $51.2_{\pm0.5}$ | $65.4_{\pm0.1}$ | $83.2_{\pm0.1}$ | $69.4_{\pm0.3}$ |
| | MaxP | $89.4_{\pm0.1}$ | $90.2_{\pm0.2}$ | $52.1_{\pm0.4}$ | $65.4_{\pm0.1}$ | $85.7_{\pm0.1}$ | $73.2_{\pm0.3}$ |

