# OpenReview forum: "Are Uncertainty Quantification Capabilities of Evidential Deep Learning a Mirage?"
_NeurIPS.cc/2024/Conference — NeurIPS 2024 poster_

### Official Review · Reviewer_cF7x · 2024-06-21

**Soundness:** 2
**Presentation:** 2
**Contribution:** 3
**Rating:** 5
**Confidence:** 4

**Summary:**

This paper highlights problems with the Evidential Deep Learning (EDL) framework, analyzes these problems, and proposes possible solutions for some of them. The paper provides a new taxonomy and a unifying objective function for a wide range of EDL methods. The authors identify that aleatoric and epistemic uncertainty estimates of EDL methods do not conform to their textbook definitions and provide a bagging distillation technique that improves the learned epistemic uncertainty.

**Strengths:**

- The range of EDL methods considered in theoretical arguments and empirical experiments is extensive which allows for convincing conclusions.
- Proving properties of a novel unified objective function is a nice way to make general claims about multiple popular EDL methods.
- The introduced taxonomy of EDL methods is easy to understand and extend.
- The OOD datasets considered in the OOD detection benchmarks are diverse and lead to OOD tasks with varying characteristics. The average score describes OOD detection capabilities well.
- The mathematical argumentations are substantial and clear.

**Weaknesses:**

- The paper has many typos and grammatical errors.
- "Second, the VI loss in Eq. (3) and UCE loss Eq. (4) turn out to be equivalent to the RPriorNet objective [...]" While the RPriorNet method uses a held-out out-of-distribution (OOD) dataset and incorporates a positive $\gamma_\text{ood}$ scalar, the VI and regularized UCE objectives only utilize an in-distribution (ID)  dataset during training. This is an essential difference between the two method classes that does not allow such a conclusion. However, all three methods are, indeed, special cases of the unified objective function in Eq. (6), utilizing different parts of it.
- The distinguishing features (2-4) discussed in L180-181 are already discussed in the referenced survey of Ulmer et al. (2023; TMLR) (e.g., in Tables 1-2 for the use of OOD data).
- The fact that epistemic uncertainty does not vanish in EDL methods is not surprising, as, considering a constant regularization factor $\lambda$, their explicit regularizer that drives them to a flat Dirichlet distribution trades off with the (deterministic) training signal from the labels. Therefore, the global minimizer will not be Bayes optimal even in the limit of infinite data. However, considering a regularization factor $\lambda$ (or, equivalently, a scaling factor $\nu = \lambda^{-1}$) that depends on the dataset size readily alleviates this problem. It also fixes the misalignment with the "dictionary definition" of aleatoric uncertainty. Based on these observations, the proof shows that keeping the regularization strength constant is a bad idea, but does not identify problems with the EDL framework. This conclusion is well known, however: in Bayesian learning, the effect of the prior also vanishes as more data is observed, and the prior can be viewed as a regularizer.
- Similarly, the fact that in the empirical results the aleatoric uncertainty is not constant is expected: estimates derived from the model are always inherently model- and objective-dependent. This holds not only for EDL methods but also for any learned uncertainty estimator.
- For many practical tasks (such as OOD detection, a widely used proxy task for epistemic uncertainty, or correctness prediction, a predictive uncertainty benchmark), the absolute scale of the epistemic uncertainty value does not matter. In particular, an estimator can be performant without being constrained to vanish in the limit of infinite data, e.g., if it reaches a strictly positive minimum in this limit. The observation of non-vanishing epistemic uncertainty does not affect the practical use of these uncertainty estimates. The authors do not discuss such practical implications, even though they use the AUROC metric for OOD detection evaluation which is scale- and shift-variant by design.
- L242 contradicts with L207. According to L207, the objective enforces $p_\psi(y \mid x)$ to fit $\frac{\mathbf{1}_C + \nu\boldsymbol{\eta}(x)}{C + \nu\mathbf{1}_C^\top\boldsymbol{\eta}(x)} \ne \boldsymbol{\eta}(x)$. This, in turn, makes the correspondence between L250 and L207 weaker.
- In the abstract, the following is stated: "[we] reveal that the EDL methods can be better interpreted as an out-of-distribution detection algorithm based on energy-based-models". This claim is unsubstantiated, as the parallel with energy-based models is only drawn for EDL approaches that utilize ID + OOD mixture datasets for training. Having access to different training signals for ID and OOD samples naturally makes the learned models better OOD detectors (c.f. Figure 5). However, as Charpentier et al. (2020; NeurIPS) point out, this assumption is unrealistic and fails to cover _all_ meaningful OOD regions. In particular, eleven out of the nineteen methods Ulmer et al. (2023; TMLR) consider do not require any form of OOD data.
- The paragraph at L304 suggests that incorporating model uncertainty would result in the "dictionary definition" of distributional uncertainty (which remains to be provided). This claim is not backed theoretically and is only supported empirically via distillation. The success of distillation might be mostly explained by the use of model ensembles (or MC-Dropout) and not the framework of Eq. (6) used for theoretical reasoning by the authors. The bridge between the theoretical argumentations and the empirical results L304 foreshadows is missing.
- The improvement upon ensembling by further bootstrapping the training datasets is marginal. Due to the lack of error bars, Figure 3 can only be used to judge the result of a single seed, but the error bars of ensembles and bootstrapped ensembles might overlap significantly. The novelty of the use of bootstrapping is only in the context of EDL: Lakshminarayanan et al. (2017; NIPS) already considered bagged ensembles (and found that non-bagged variants performed better).
- In the empirical evaluation several standard evaluation metrics are missing, e.g., the ECE score and (strictly) proper scoring rules. While total uncertainty and predictive uncertainty are evaluated, the aleatoric uncertainty estimate is not benchmarked separately. This would be key to back the claim that EDL methods are (mostly) OOD detectors: perhaps they are just as aligned with aleatoric uncertainty as epistemic, which would refute this claim.
- The authors do not benchmark against a baseline, e.g., a cross-entropy-trained deterministic network. This makes the connection of results with methods outside the EDL framework hard.

**Minor issues:**
- "underlying distribution" appears twice in L79, making the sentence hard to parse.
- $D(p\ \Vert\ q)$ is an ambiguous notation for the Kullback-Leibler divergence. For example, it is often used for general Bregman divergences. Consider using the notation $D_\text{KL}(p\ \Vert\ q)$.
- $h(\cdot)$ is also an unusual notation for entropy. $\mathbb{H}(\cdot)$ or $\mathrm{H}(\cdot)$ are the usually used notations.
- The UCE loss was not proposed by Charpentier et al. (2020, NeurIPS). Their contribution was the Regularized UCE. UCE was proposed by Biloš et al. (2019, NeurIPS). Therefore, the notation $\ell_\text{UCE}$ is also questionable.
- "Other choices of prior were also proposed to promote some other desired property." Such sentences are too ambiguous for an academic text. Consider naming (at least some of) the desired properties before pointing to further resources.

**Questions:**

- "We use validation accuracy as the criterion to save the optimal model, as we observe using validation loss leads to unstable results." What does "unstable results" mean here? Additionally, have the authors tried to choose the optimal model based on their evaluation criteria (e.g., the selective classification performance)?
- Does bootstrapping lead to a qualitatively different uncertainty quantification behavior compared to regular ensemble training? This is hard to judge, as Fig. 10 only features Bootstrap.
- The extrapolated epistemic uncertainties of methods not utilizing OOD data are seemingly random based on Fig. 5. What might be the reason for the high AUROC score for most of these methods in Fig. 3? In particular, might it be the case that the OOD datasets are so different from their ID variants that the task becomes easy even with ill-posed epistemic uncertainties?

**Limitations:**

The authors do not adequately discuss their work's limitations, as they only acknowledge that their unified objective function does not cover all EDL methods and that their Bootstrap Distillation method is costly.
- Several standard evaluation metrics are missing from the empirical evaluation.
- The aleatoric uncertainty estimates are not evaluated quantitatively.
- The paper does not present the performance of a simple baseline (e.g., a deterministic neural network's predictive entropy), making the difficulty of the tasks hard to estimate.
- The authors do not formally argue why incorporating model uncertainty would alleviate the problems with aleatoric and epistemic uncertainty estimation outlined in Section 5.1.

---

> ### Author Rebuttal · Authors · 2024-08-07
>
> We thank the reviewer for their thoughtful comments and address them in detail below.
>
> **1. The difference between RPriorNet and VI, UCE cannot support the claim " the VI loss in Eq. (3) and UCE loss Eq. (4) turn out to be equivalent to the RPriorNet objective [...]"**
>
> We acknowledge that this sentence could be misleading, and we will revise it accordingly. What we want to convey is the resemblance in the in-distribution (ID) part of the objectives. Specifically, although VI loss and UCE loss were originally derived from different motivations, they are equivalent to the ID part of the VI objective in disguise. To our knowledge, this observation has not been made before, and we believe it provides a unified way to reason about a class of EDL methods in the literature.
>
> **2. EDL’s non-vanishing epistemic uncertainty is not surprising, and a data size dependent $\lambda$ can resolve this issue.**
>
> We agree that epistemic uncertainty would vanish with a choice of data size dependent $\lambda$’s. However, we would like to point out a few key considerations. First, the classical Bayesian approach does not require a data size-dependent prior. As long as the data size goes to infinity, the epistemic uncertainty will eventually vanish regardless of the choice of prior. Second, it remains unclear how to properly define the data size-dependent $\lambda$, in particular, its decay rate. Third, while it is true that one can always propose heuristic tricks to make the uncertainty of any algorithm align with the dictionary definition, this does not imply that such an algorithm is a faithful uncertainty quantifier.
>
> **3. EDL’s non-constant aleatoric uncertainty is expected, this holds for any uncertainty estimator.**
>
> Although it might be correct to say that aleatoric uncertainty is implicitly dependent on the model's capacity, randomness in training, data sampling, and other artifacts, what we are trying to argue is that, even under ideal scenarios (infinite sample size, overparameterized model), aleatoric uncertainty quantified by EDL methods can still be dependent on hyper-parameters ($\lambda$).
>
> **4. EDL’s non-vanishing epistemic uncertainty does not affect the practical use of this uncertainty in downstream tasks.**
>
> First, in this paper, we do not claim that EDL methods are ineffective for downstream tasks. Instead, we attempt to answer the following question: “Why and how do EDL methods work in practice?” As a first step, we characterize the global optimizer of several EDL methods and highlight several consequences, including the non-vanishing epistemic uncertainty. We believe that a clear understanding of the objectives used for "uncertainty quantification" is crucial to understanding how EDL methods actually behave, how to further improve these methods, and how to lead to a more reliable and grounded UQ approach.
>
> Second, while we agree that non-vanishing epistemic uncertainty might not be directly related to common UQ downstream tasks such as OOD detection, there are practical applications that require accurate quantification of epistemic uncertainty. For example, in active learning, we cannot rely on the EDL model’s epistemic uncertainty as a signal to indicate the model’s actual knowledge at a certain stage because it never vanishes. The $\lambda$-dependent aleatoric uncertainty also may have issues in certain applications. For example, suppose an application relies on multiple pre-trained models to make predictions and identify ambiguous samples. If the learned aleatoric uncertainty of each model is trained with different $\lambda$'s in an EDL framework, then they might not be in a comparable scale.
>
> **5. L242 contradicts with L207.**
>
> In the paragraph, we consider the standard regime where $\nu$ is sufficiently large, so that $\alpha_0+\nu\eta(x)$ is approximately $\nu\eta(x)$, and the correspondence approximately holds in that regime.
>
> **6. The claim “the EDL methods can be better interpreted as an out-of-distribution detection algorithm based on energy-based-models”' is unsubstantiated, as multiple EDL methods do not require any form of OOD data.**
>
> We acknowledge that such resemblance applies to not all EDL methods, but to the methods covered by the unified objective (6). The main claim of this section is that the unified objective (6) closely resembles the EBM-based OOD detection objective. When no OOD data is available, the resemblance still remains, if we ignore the OOD regularization term. In that case, they simply promote very large outputs for ID data.
>
> The empirical success of EDL methods that do not assume OOD data, such as PostNet, implies that OOD data may not be required for OOD detectors to distinguish OOD data for some reason. We conjecture that this success is an artifact due to a combination of some optimization phenomena and neural networks’ extrapolation ability. In fact, similar to the eleven out of the nineteen methods in Ulmer et al., there are also many effective OOD detection algorithms in the literature that do not require any OOD data during training.
> We acknowledge that this point was not made clear in the submission, and we will revise the section accordingly, especially in the case when no OOD data is available.
>
> **7. The claim about the benefit of incorporating model uncertainty is not backed theoretically.**
>
> Please refer to global response 1.
>
> **8. Missing standard evaluation metrics such as ECE.**
>
> Please refer to global response 2.(1) and Figure 13.
>
> **9. Missing aleatoric uncertainty benchmark.**
>
> Please refer to global response 2.(2) and Figure 14., 15.
>
> **10. Missing benchmark against a cross-entropy-trained deterministic network.**
>
> Please refer to global response 2.(3).

---

> ### Author Response · Authors · 2024-08-07
> **Rebuttal by Authors (Part II)**
>
> **11. L180-181 are already discussed in the referenced survey of Ulmer et al. (2023; TMLR).**
>
> We acknowledge that lines 180-181 are also discussed in Ulmer et al., but these were not used to classify the EDL methods in Ulmer et al. We argue that the features mentioned in lines 180-181 can better classify the EDL methods with our unified view on the objective functions, and the dichotomy between “PriorNet-type methods” and “PostNet-type methods” in Ulmer et al. may not effectively contrast different EDL methods compared to our taxonomy. We will clarify our arguments accordingly in the revision.
>
> **12. The improvement upon ensemble distillation is marginal; The novelty of the use of bootstrapping is only in the context of EDL.**
>
> First, the detailed numbers with standard error are provided in Tables 2, 3, and 4 in the Appendix, which shows that the improvement is statistically significant. Second, we did not claim that bootstrap distillation is superior to ensemble distillation. Instead, we aim to advocate for distillation-based methods in general: ensemble distillation is the existing algorithm, and we propose bootstrap distillation as yet another alternative. Third, the main contribution of this work is to offer an analysis to better understand how EDL methods behave and provide insights on how to improve them. Proposing a novel EDL algorithm that achieves SOTA downstream task performance is beyond the scope of this work and could be considered a separate publication.
>
> **13. Question 1.**
>
> For RPriorNet, which utilizes OOD data for training, both the training loss and validation loss can fluctuate significantly due to the conflict between in-distribution and OOD data optimization. In this case, validation accuracy is a better signal to ensure the saved model is well-optimized. For other EDL methods, we rely on validation loss and never use any downstream task performance to choose the optimal model
>
> **14. Question 2.**
>
> We note that bootstrap distillation and ensemble distillation only differ in terms of the underlying stochastic algorithm, and the induced behavior depends on the choice of stochasticity.
>
> **15. Question 3.**
>
> First, as we mentioned in response 8, we conjecture that the success of EDL methods without OOD data is an artifact due to a combination of certain optimization phenomena and the extrapolation ability of neural networks. Second, the success is relative to other EDL methods, which actually suggests that other EDL methods relying on auxiliary techniques are not reliable for real data tasks (see Section 5.3).
>
> If these clarifications satisfactorily address the reviewer's concerns, we kindly ask if the reviewer would consider updating the score to reflect what we believe is a paper with noteworthy contributions to the community.

---

> ### Comment · Reviewer_cF7x · 2024-08-09
>
> Dear Authors, thank you for your rebuttal.
>
> **The difference between RPriorNet and VI, UCE cannot support the claim " the VI loss in Eq. (3) and UCE loss Eq. (4) turn out to be equivalent to the RPriorNet objective [...]"**
>
> I understand the aim to highlight the resemblance in the ID part of the objectives. However, the paper neglects the importance of including or excluding the OOD data from the training objective. This choice is a key characteristic of EDL methods, thus saying that the "VI loss [...] and UCE loss [...] turn out to be equivalent to the RPriorNet objective [...]" is misleading.
>
> **EDL’s non-vanishing epistemic uncertainty is not surprising, and a data size dependent λ can resolve this issue.**
>
> > the classical Bayesian approach does not require a data size-dependent prior
>
> Consider a Bayesian maximum a posteriori learning task using a Gaussian isotropic prior. Then $- \log p(\theta \mid \mathcal{D}) \equiv -\sum_{i=1}^n\log p(y_i \mid x_i, \theta) + \frac{\tau}{2}\Vert \theta \Vert_2^2$ where $\tau$ is the prior precision. Further dividing by $n$ results in an equivalent objective: this now matches Eq. (6) with the first term being an expectation and the second term being $\frac{\tau}{2n}\Vert \theta \Vert_2^2$, which depends on $n$.
>
> > As long as the data size goes to infinity, the epistemic uncertainty will eventually vanish regardless of the choice of prior.
>
> The same holds for the EDL objective with a data-dependent $\lambda$. The cited challenges of scheduling the decay of $\lambda$ only affect the theoretical argumentation for the well-behavedness of the objective.
>
> **EDL’s non-vanishing epistemic uncertainty does not affect the practical use of this uncertainty in downstream tasks.**
>
> The paper does not sufficiently investigate the question “Why and how do EDL methods work in practice?”. Investigating the global minimizer of a unified objective is indeed useful, but the consequences the authors enumerate do not notably affect the practical usability of EDL methods. In particular, active learning also requires an effective _ranking_ of inputs based on the importance of acquiring supervision on them. Shifting the uncertainty values by a constant does not affect this ranking.
>
> **The claim “the EDL methods can be better interpreted as an out-of-distribution detection algorithm based on energy-based-models” is unsubstantiated, as multiple EDL methods do not require any form of OOD data.**
>
> > We acknowledge that such resemblance applies to not all EDL methods, but to the methods covered by the unified objective (6).
>
> The resemblance applies only to EDL methods that leverage OOD data during learning. There is no such resemblance to EDL methods that only utilize ID data: "if we ignore the OOD regularization term", the learning dynamics become substantially different. Please refer to the first point of this reply.
>
> **The improvement upon ensemble distillation is marginal; The novelty of the use of bootstrapping is only in the context of EDL.**
>
> Consider including the error bars in the bar plots as well.
>
> **Additional experiments.**
>
> Thank you for adding the requested experiments. They clarify the behavior of different EDL methods and allow for easier comparability.
>
> **Conclusion.**
>
> The authors have addressed some of my concerns. In light of their clarifications, I am willing to increase my score from 4 to 5. However, due to the remaining points discussed above and the work's limited impact, I do not propose a higher score.

---

> > ### Author Response · Authors · 2024-08-10
> > **Request for updating the visibility**
> >
> > We apologize for the inconvenience, but **could you update the visibility of your response to include `Reviewers Submitted`?**
> > We suspect that, as our Rebuttal Part II was posted as "official comment" and the system didn't allow it to be visible to other reviewers at the time of submission, your comment inherited the limited visibility. We have updated the visibility of our Part II, so you should be able to update it too. We believe that making our discussion visible to other reviewers would help other reviewers. Sorry again for the inconvenience.

---

> ### Author Response · Authors · 2024-08-10
>
> $$\newcommand{\piv}{\boldsymbol{\pi}}\newcommand{\av}{\boldsymbol{\alpha}}\newcommand{\bv}{\boldsymbol{\beta}}\newcommand{\E}{\mathbb{E}}$$
> We thank the reviewer for engaging in the discussion and sharing additional comments.
> We are glad that some of our concerns have been addressed, and we appreciate the reviewer for raising the score.
>
> Here, we wish to clarify concerns in some of the additional comments as follows.
>
> **1. Regarding the resemblance between EDL method without OOD data and EBM-based OOD detection algorithm.**
>
> The reviewer is correct that EDL methods without OOD data are not exactly equivalent to the specific method proposed by [24], which explicitly assumes OOD data during training. However, we wish to clarify that **we can still argue a close resemblance if we ignore the OOD regularization in both objectives** as follows:
>
> If we *remove* the OOD-data-dependent term in the EBM-based OOD detector’s objective, the objective becomes
> $$-\E\_{p(x,y)}[\log {p_\phi(y|x)}] + \tau \E\_{p(x)}[\max(0,E_\phi(x)-m_{\textsf{id}})^2],$$
> where $E\_\phi(x)= -\log \boldsymbol{1}\_C^\intercal\bv\_\phi(x)$.
>
> This still has a similar effect to the ID part of the unified objective in eqn. (6), which is in turn simplified in eqn. (8) as
> $$
> \E\_{p(x)}[D(\mathsf{Dir}(\piv;\av\_\psi(x)) \~\|\|\~ \mathsf{Dir}(\piv; {\av\_0+\nu\boldsymbol{\eta}(x)}))].
> $$
> That is, both objectives promote $\bv\_\phi(x)$ and $\av\_\psi(x)$ to output large values for ID data, while to be aligned with the underlying label distribution $\boldsymbol{\eta}(x)=(p(y|x))_{y=1,\ldots,C}$ when normalized, provided that $\nu \gg 1$.
>
> We will clarify these points in the revision of the manuscript.
>
> **2. Data size-dependent prior in Bayesian.**
>
> The example provided by the reviewer, in fact, suggests that $\frac{1}{N}$ term in the Bayesian framework is naturally induced, rather than heuristically crafted. Here, the $\theta$ (model parameter) in Bayesian framework is a “global” random variable dependent on all the samples in dataset $\mathcal{D}$, when the model observes more samples, the prior belief (of model parameter) is naturally dominated by likelihood, and the posterior become more concentrated.
>
> However, EDL methods do not have such a nice property. The reason is straightforward:
> In the EDL framework, $\pi$ is a “local” variable dependent on each sample $x$ instead of shared by all samples in the datasets,  and its uncertainty is induced by its unique second order distribution $p_{\psi}(\piv|x)$ (parameterized by network $\psi$). To make the same analogy as the Bayesian example provided by the reviewer, we need multiple labels for each $x$, i.e., $\mathcal{D}\_x = \\{(x, y\_1), (x, y\_2), \ldots, (x, y\_{N\_x})\\}$, where $y_1, y_2, \dots, y_{N_x}$ are $N_x$ samples from the underlying first-order label distribution $p(y|x)$.
> Then, we could end up with a similar objective of the form
> $$-\log p(\piv|\mathcal{D}\_x) = - \sum\_{i=1}^{N} p(y\_i|\piv) + \lambda (\text{prior\\_term}).$$
>
> In practice, however, for each $x$, we only observe a single label $y$, as $x$ is continuous and/or extremely high dimensional like images, and thus, in the EDL methods, the effect of prior does not naturally vanish as the sample size increases.
> A similar intuition has also been offered in some recent works, e.g., see the third paragraph of Section 3.3 in [4] and the second paragraph of Section 6 in [5].
>
> - **On the practical usage of EDL methods**
>
> We agree that `active learning also requires an effective ranking of inputs based on the importance of acquiring supervision on them. Shifting the uncertainty values by a constant does not affect this ranking`.
> We wish to remark, however, that beyond the query strategy (such as ranking), another important problem in active learning is to decide when to stop querying new labeled samples to minimize the labeling budget.
> It is unclear whether a heuristic trick such as "stopping data acquisition as long as the uncertainty becomes a constant" is reliable, because if the model queries bad samples in the previous iterations or the model is not well-trained, its uncertainty can also be roughly constant compared to previous iterations without improvement. Stopping the model training in such scenarios is not desirable. Uncertainty learned with a relative scale would render such a decision even more difficult.
>
> Ideally, we believe that uncertainty accurately quantified in an absolute scale can better reflect how a machine learning system is truly lacking knowledge, especially for high-stake applications, such as medical diagnosis, and finance, where users need to rely on uncertainty measures to make important decisions.
>
> **References**:
> - [24] Energy-based Out-of-distribution Detection, Liu et al., NeurIPS 2020.
> - [4] Pitfalls of Epistemic Uncertainty Quantification through Loss Minimisation, Bengs et al., NeurIPS 2022.
> - [5] On Second-Order Scoring Rules for Epistemic Uncertainty Quantification, Bengs et al., ICML 2023.

---

> > ### Comment · Reviewer_cF7x · 2024-08-11
> >
> > Dear Authors,
> >
> > Thank you for your answer. Your arguments have not resolved my remaining concerns; therefore, I retain my score.

---

> > > ### Author Response · Authors · 2024-08-13
> > >
> > > We appreciate your reply and engagement in the discussion. Could you kindly elaborate on the concerns that remain and how we can address them? We are eager to improve the quality of the paper, and your feedback would be greatly valuable.

---

> > > > ### Comment · Reviewer_cF7x · 2024-08-13
> > > >
> > > > Dear Authors,
> > > >
> > > > Thank you for your reply.
> > > >
> > > > **1. Regarding the resemblance between EDL methods without OOD data and EBM-based OOD detection algorithms.**
> > > >
> > > > Please refer to my earlier comment. Ignoring the OOD regularization in both objectives changes the learning dynamics substantially. The method of Liu et al. uses OOD data -- ignoring the OOD term does not allow the authors to draw a parallel between EDL methods utilizing only ID data and the objective used by Liu et al. The fact that the in-distribution terms "have a similar effect" is a rather weak statement.
> > > >
> > > > **2. Data size-dependent prior in Bayesian learning.**
> > > >
> > > > The authors seem to have misunderstood the example I provided. The example highlights the inaccuracy of the claim that "the classical Bayesian approach does not require a data size-dependent prior." Dividing the negative log-likelihood objective by the number of training samples results in an equivalent objective, whose terms can now be matched to the terms of Eqs. (2-4) (assuming a loss over multiple inputs). While the prior itself is not data-size-dependent, the induced regularization strength _does_ depend on the number of training samples in Bayesian learning. Choosing a data-size-dependent $\lambda$ in the paper's analysis resolves the erratic limiting behavior of Eq. (6), which the authors also acknowledge ("We agree that epistemic uncertainty would vanish with a choice of data size dependent $\lambda$’s"). Theorem 5.1 and Example 5.2 show that choosing a constant regularization strength independent of the dataset size leads to incorrect limiting behavior. However, it is a widely known fact that the optimal regularization strength in a regularized risk minimization problem _does_ depend on the dataset size. Therefore, I consider the impact of this observation limited.
> > > >
> > > > **3. On the practical usage of EDL methods.**
> > > >
> > > > Stopping the active learning loop based on the performance of the learned predictor alleviates the problem with the heuristic approach detailed by the authors. I would like to point out that a not well-trained model can be confidently incorrect, and stopping criteria that only take the model's uncertainty and not its performance into account can lead to premature stopping even when the absolute scale of the uncertainty estimates is known. Further practical tasks that are agnostic to this absolute scale are (binary) OOD detection, correctness prediction, selective prediction, or corrupted input detection.

---

> > > > > ### Author Response · Authors · 2024-08-13
> > > > >
> > > > > We appreciate the reviewer's prompt and detailed response. We will discuss these points appropriately in the revision to clearly delineate the implications and limitations of our analyses and observations.

---

### Official Review · Reviewer_gUpP · 2024-06-28

**Soundness:** 3
**Presentation:** 2
**Contribution:** 3
**Rating:** 6
**Confidence:** 3

**Summary:**

The presented paper offers a (further) critique on EDL. They show that a range of EDL objective functions are largely equivalent by presenting a unified objective function that subsumes many existing objective functions. With this unified objective function some problematic properties are shown that prove that these EDL methods cannot properly represent aleatoric and epistemic uncertainty. Following this, they show various failure modes of EDL. Additionally, they show that observed good performance on OOD-detection is not robust and may not work well with even moderate dimensionality. Lastly, they say that distillation-based methods are a better alternative and demonstrate an adaptation to a distillation-based method that outperforms existing methods.

**Strengths:**

- The authors expand upon an existing critique by showing that many seemingly different EDL methods are actually very similar, collapsing multiple different methods into a single conceptual understanding (originality).
- Some of the practical weaknesses shown of EDL also appear to be novel, such as unchanged epistemic uncertainty under different dataset sizes (significance).
- The claims surrounding the critique of EDL appear to be well supported and sound (quality)
- The observation that EDL methods for OOD detection can be sensitive to changes in architecture and that it may fail in moderately high dimensionality give suggestions on how OOD detection using density estimation generally may or may not work. This may have implications outside of EDL methods, and generally says something about how Neural Networks learn. (significance)

**Weaknesses:**

- The central point of critiquing EDL is not novel. The authors acknowledge that and do progress the critique, but I do think doubling down on the critique may have limited impact (originality)
- The investigation into the newly proposed Bootstrap distillation is section 7 is lacking. Leave one-class-out OOD detection is not considered. The observation that the predicted epistemic uncertainty is monotonically increasing with accuracy is not sufficient to claim that it is “faithfully representing” epistemic uncertainty. (quality)
- Some parts of the critique are about EDL’s failure in estimating epistemic uncertainty. However, the claim that EDL’s “distributional uncertainty” is “a kind of epistemic uncertainty” does not seem to have much support. Theoretically that claim is not sound (as epistemic uncertainty is the uncertainty in the model parameters), and the authors provide no sources that make this claim. Without a good source of why EDL is expected to represent epistemic uncertainty the critique on its epistemic uncertainty is meaningless. I was able to find some papers that make this claim: https://proceedings.neurips.cc/paper/2018/file/a981f2b708044d6fb4a71a1463242520-Paper.pdf, https://openreview.net/pdf?id=UI4K-I2ypG. Consider adding some discussion on the (mistaken) belief in the field that EDLs distributional uncertainty represents epistemic uncertainty. (clarity/quality)
- The density of the paper (many experiments and findings for only 9 pages) means a large part of the methods and results are pushed to the appendix. As a result, the implementation details are not always clear. How distillation-based methods work is not sufficiently explained. (clarity)
- The newly proposed bootstrap-distillation method is unlikely to have a meaningful impact considering it’s limited theoretical argumentation and the limitations of that method are minimally explored. (significance)

**Questions:**

- The bootstrap-distillation method seems to have limited relation to the current work. Distillation methods in general seem relevant, but the newly proposed bootstrapping method compared to END^2 seems to be unnecessary and not completely discussed. Would the authors consider introducing the bootstrap-distillation method as a separate publication (with space for more theory and more experiments)? I think this would also allow some of the more important parts of the Appendix to be moved into the main body of the current work.
- The demonstration in Fig 1.a is a bit confusing. How do you define epistemic uncertainty in the EDL? E.g. PriorNet discusses “distributional uncertainty”, but does not consider this to be equivalent to epistemic uncertainty. Can you cite any sources that claim that the distributional uncertainty from EDL should be equivalent to epistemic uncertainty? And ideally that this kind of behavior of being “reducible” with more data should follow?
- Line 105 says “distributional uncertainty” is a kind of “epistemic uncertainty”. What is this claim based on? Is there a source for this? It seems that p(psi|D) would still be the epistemic uncertainty in this decomposition (which is reduced to a single point).
- It is correct that lambda affecting the predicted aleatoric uncertainty is problematic. However, what practical problems will this cause (and will this not cause) when aleatoric uncertainty is considered for downstream tasks? Identifying difficult, ambiguous or mislabelled samples may still be perfectly functional.

**Limitations:**

EDL objective functions that are not subsumed by the unified objective function are discussed.

The authors give a thorough critique of the failures of EDL, with the implication that Bayesian NNs would be the better alternative. However, they are not explicitly compared. Adding the behavior of a BNN in Figures 1.a, 4.a and 10 would demonstrate whether EDL are also worse than BNN-based methods. This is a limitation and it is not discussed.

---

> ### Author Rebuttal · Authors · 2024-08-07
>
> We thank the reviewer for their thoughtful comments. We address them in detail below.
>
> **1. The central point of critiquing EDL is not novel.**
>
> While novelty is in the eye of the beholder, and critiquing EDL is indeed a primary focus of this work, our contribution is non-trivial compared to existing papers. First, we offer a more advanced understanding of EDL methods compared to existing critiques [4, 5, 6] (see Appendix A). Second, beyond the critique, this work is the first to attempt to remedy the limitations of EDL methods. Third, to the best of our knowledge, this work is also the first to unify multiple representative EDL methods into a single framework for theoretical analysis, supported by empirical justification with real data experiments.
>
> **2. Regarding the support and the source of the claim “distributional uncertainty is a kind of epistemic uncertainty”.**
>
> We respectfully point out that this is a common claim in the EDL literature. We provide a list of supporting evidences from the literature as follows:
> - Malinin et al. (NeurIPS 2018) [7] proposed the "forward Prior Network". In this work, they introduced the concept of "distributional uncertainty," which is different from "model uncertainty" in Bayesian methods. However, similar to "model uncertainty," "distributional uncertainty" also measures the model's lack of knowledge, so it is aligned with the definition of epistemic uncertainty. They mentioned, “Distributional uncertainty is an ’unknown-unknown’ - the model is unfamiliar with the test data and thus cannot confidently make predictions.” The follow-up work of the same group of authors Malinin et al. (NeurIPS 2019) [8] also noted that “Knowledge Uncertainty, also known as epistemic uncertainty or distributional uncertainty, arises due to the model’s lack of understanding or knowledge about the input.”
> - Malinin et al. (ICLR 2020) [9] proposed the distillation-based method “END^2”. Therein, the authors again remarked that “Knowledge uncertainty, also known as epistemic uncertainty, or distributional uncertainty, is uncertainty due to a lack of understanding or knowledge on the part of the model regarding the current input for which the model is making a prediction.”
> - Charpentier et al. (NeurIPS 2020) [15], where the“Posterior Network” was proposed, uses "epistemic uncertainty" to refer to the "distributional uncertainty." The same usage can be found in its follow-up work (Charpentier et al., ICLR 2022) [20], in which “Natural Posterior Network” was proposed. There is no mention of “distributional uncertainty” in both works.
> - Recently, several critiques of the EDL methods such as Bengs et al. (NeurIPS 2022) [4], Bengs et al. (ICML 2022) [5], Jürgens et al. (ICML 2024) [6] have adopted the same convention. In these papers, they all use “epistemic uncertainty” as the terminology to refer to EDL's “distributional uncertainty” and criticize that EDL cannot faithfully quantify epistemic uncertainty.
>
> We will include these sources in the manuscript to avoid any confusion.
>
> **3. Regarding paper presentation and clarity.**
>
> We acknowledge the content is a bit cluttered in the submission due to the page limit. In the revised manuscript, we will do our best to improve clarity and elaborate on the details with the additional page.
>
> **4. Lacking theoretical justification and further investigation of proposed Bootstrap distillation.**
>
> Please refer to global response 1.
>
> **5. Would the authors consider introducing the bootstrap-distillation method as a separate publication?**
>
> The main point of this work is to advocate for distillation-based EDL methods in general, and the proposed bootstrap-distillation method serves as another alternative in the ensemble approach to justify this argument. We agree with the reviewer that conducting a comprehensive analysis (both theoretical and empirical) of distillation-based EDL methods can be an interesting future work.
>
> **6. What practical problems will $\lambda$ dependent aleatoric uncertainty cause in downstream tasks?**
>
> It is correct that identifying difficult, ambiguous, or mislabelled samples is still possible in practice, as long as the downstream task only relies on a relative scale of uncertainty values. However, there might be practical scenarios where the absolute value of uncertainty also matters. For example, suppose an application relies on multiple pre-trained models to make predictions and identify ambiguous samples. If the learned aleatoric uncertainty of each model is trained with different $\lambda$'s in an EDL framework, then they might not be in a comparable scale, which prohibits any further comparison.
>
>
> **7. Missing the behavior of a Bayesian approach in Figures 1.a, 4.a and 10.**
>
> Please refer to global response 2.4.

---

> > ### Comment · Reviewer_gUpP · 2024-08-08
> >
> > **1. The central point of critiquing EDL is not novel.**
> > > this work is the first to attempt to remedy the limitations of EDL methods
> >
> > The bootstrap-distillation is this first attempt. However, it’s rather weakly analysed. Effectively it’s a combination of BNN-based methods and EDL based methods but should then be compared to both extensively.
> >
> > > this work is also the first to unify multiple representative EDL methods into a single framework
> >
> > I acknowledge that this is a substantial contribution. I think it is interesting and shows insight into the various EDL methods and how they relate. However, this is a nice insight for a small subfield of EDL. I think this is why a 6 is fitting.
> >
> > **2. Regarding the support and the source of the claim “distributional uncertainty is a kind of epistemic uncertainty”.**
> >
> > I acknowledge that this (confused) claim is indeed somewhat common. I believe adding the proposed references will strengthen the argument.
> >
> > **4. Lacking theoretical justification and further investigation of proposed Bootstrap distillation.**
> >
> > The additional results are beneficial, but don’t constitute a substantial analysis comparing the Bootstrap Distillation with BNN-based methods and EDL based methods.
> >
> > **7. Missing the behavior of a Bayesian approach in Figures 1.a, 4.a and 10.**
> >
> > Thank you for adding these results. I agree that the behavior of the ensemble has been extensively studied but I think adding them to the manuscript gives a clear comparison.
> >
> >
> > ## Conclusion
> >
> > I appreciate the authors engaging in the discussion. I agree that unifying the existing EDL approaches is interesting, but I doubt it is impactful. The same applies to the further investigation into EDL failure cases. The Bootstrap Distillation could be more impactful, but the current analysis remains not sufficient to show this. I stay at a “Weak accept” evaluation.

---

> > > ### Author Response · Authors · 2024-08-12
> > >
> > > We thank the reviewer for appreciating the contribution of this work, and providing valuable comments. We will carefully revise the manuscript to incorporate the suggestions.

---

### Official Review · Reviewer_LBKP · 2024-06-30

**Soundness:** 3
**Presentation:** 2
**Contribution:** 2
**Rating:** 5
**Confidence:** 4

**Summary:**

The paper focuses on Evidential Deep Learning (EDL) models developed for uncertainty quantification in a computationally efficient manner. It identifies key limitations of the EDL methods: their inability to faithfully express both the epistemic and aleatoric uncertainties. The paper then proposes to integrate model uncertainty into the EDL-based models to address the limitations of existing EDL models with a Bootstrap-distill method. The bootstrap-distill methods train multiple models on a subset of training data. Each model trains on a disjoint subset of the dataset to capture the model uncertainty. Experiments are carried out over multiple benchmark datasets to demonstrate the weaknesses of EDL models and the effectiveness of their proposed bootstrap-distill model.

**Strengths:**

- The paper is generally well-written and easy to follow
- The paper discusses the effectiveness of the evidential deep learning models and highlights their limitations regarding uncertainty quantification.
- The relationship of EDL methods with EBM-based OOD detectors further helps better understand the EDL based methods.
- Experiments are carried out with benchmark datasets to empirically validate their claims. The experiments justify the claims regarding the weaknesses of EDL models.

**Weaknesses:**

- The proposed solution of introducing model uncertainty p(\psi|D) over the point estimate \delta(\psi - \psi*)of the EDL approach to address the limitations seems to be a heuristic choice. Is the proposed solution theoretically guaranteed to faithfully express the epistemic/aleatoric uncertainties?
- In my understanding, EDL approach presents a computationally efficient Uncertainty Quantification (UQ) approach that is alternative to Bayesian approaches as it only involves one forward pass. However, introducing the model uncertainty p(\psi|D) would lead to the same computational expense compared to Bayesian approaches. It is not clear if there is any benefit of the p(\psi|D)-based EDL framework in UQ over Bayesian approaches for UQ.
- The proposed solution  (bootstrap-Distil)'s aleatoric/epistemic uncertainty behavior is not well justified.  Maybe, the uncertainty behavior of bootstrap could be added in Figure 1/4 to better justify the proposed model. Also Figure 10: Which dataset is used? How is the epistemic uncertainty quantified for the bootstrap method? How does it compare with standard UQ methods (eg. bayesian ensembling and dropout uncertainty). Also, the range of epistemic uncertainty seems to be small (0.1 - 0.0001). Is there any range of the aleatoric/epistemic uncertainty values? Are the uncertainty values relative, or is there any meaning to the values?
- Comparison with SOTA OOD detection methods is missing. Only EDL-based models are considered where the proposed model is superior.
- Experiments and results are shown on simple datasets such as Cifar10 and Cifar100. It would be interesting to see the developed model's uncertainty results along with baseline model's results for more challenging datasets such as imagenet.

**Questions:**

→ Impact of hyperparameter lambda in EDL (Figure 2): I wonder how the OOD detection performance works for different evidential models for lambda = 0 (It seems as if using no regularization (\lambda = 0) might lead to the best performance.)
→ The proposed solution: Using model distribution p(\psi|D) for EDL models instead of the point estimates would lead to three levels of uncertainty: Model distribution through p(\psi|D), distributional uncertainty p(\pi|x,\psi) and aleatoric uncertainty through p(y|\pi). I wonder if there is any benefit of such a three-level framework over standard Bayesian approaches (i.e. without any distributional uncertainty)?
→ Will the proposed solution be theoretically better than the EDL methods i.e. will the method guarantee desired epistemic/aleatoric uncertainty behavior?
--> Recent works have shown learning deficiencies in evidential deep learning models[1],[2]. I wonder if the proposed model will be robust to such learning deficiencies.
[1] Learn to accumulate evidence from all training samples: theory and practice (ICML 2023)
[2] Ucertainty regularized evidential regression (AAAI 2024)
→ The claim that EDL methods perform strongly for OOD detection is not well justified. Do these EDL-based models outperform the SOTA OOD detection methods? Comparison with SOTA OOD detection methods will further strengthen the work.
--> Uncertainty behavior of the developed model (bootstrap): Is the aleatoric/epistemic uncertainty of this model accurate on all datasets and settings? How does the aleatoric uncertainty behave with an increase in sample size for the baseline and the developed model?

**Limitations:**

The work is towards understanding the limitations of EDL-based models that is well presented. The limitations of their proposed model could be further discussed.

---

> ### Author Rebuttal · Authors · 2024-08-07
>
> We thank the reviewer for their thoughtful comments. We address them in detail below.
>
> **1. Is the proposed solution theoretically guaranteed to faithfully express the epistemic/aleatoric uncertainties?**
>
> Please refer to global response 1.
>
> **2. It is unclear if there is any benefit of the $p(\psi|D)$-based EDL framework over Bayesian approaches.**
>
> The main benefit of distillation-based EDL is its computational efficiency at inference time. Once the model is trained, it does not require multiple inferences like Bayesian or ensemble methods.
>
> **3. The proposed solution (bootstrap-Distil)'s aleatoric/epistemic uncertainty behavior is not well justified.**
>
> The bootstrap-distill method does not have any hyper-parameters in its objective, so the aleatoric uncertainty is not model/hyper-parameter dependent. Regarding the behavior of epistemic uncertainty, we provide the figure of the epistemic uncertainty vs. sample size curve on the CIFAR10 dataset in Figure 10. The behavior of uncertainty quantified by bootstrap, ensemble, and Bayesian approaches is similar, as the epistemic uncertainty is induced by model uncertainty. All these methods are well-studied in literature, and they are usually considered as faithful uncertainty quantifiers.
>
> **4. Regarding the range of the aleatoric/epistemic uncertainty values. Are the uncertainty values relative, or is there any meaning to the values?**
>
> The range of aleatoric/epistemic uncertainty depends on the uncertainty metric. For example, epistemic uncertainty measured using "mutual information" will be non-negative. For typical downstream tasks such as OOD data detection, only the relative uncertainty matters for identifying OOD samples. However, there might be practical scenarios where the absolute value of uncertainty also matters. For example, in active learning, we cannot rely on the EDL model’s epistemic uncertainty as a signal to indicate the model’s actual knowledge at a certain stage because it never vanishes. The $\lambda$-dependent aleatoric uncertainty also may have issues in certain applications. For example, suppose an application relies on multiple pre-trained models to make predictions and identify ambiguous samples. If the learned aleatoric uncertainty of each model is trained with different $\lambda$'s in an EDL framework, then they might not be in a comparable scale.
>
> **5. Missing Comparison with SOTA OOD detection methods.**
>
> First, the focus of this work is NOT to propose a novel algorithm that outperforms all SOTA OOD detectors. Instead, the main contribution is to offer an analysis to better understand how a specific type of UQ method (EDL) behaves and to provide insights on how to mitigate its limitations. The bootstrap-distill approach serves as one of many possible solutions to address the issues of existing EDL methods. Furthermore, while comparing EDL methods with algorithms specifically designed for OOD detection might not be fair, we include an additional baseline comparison with a classical cross-entropy-trained network (see global response 2.3).
>
> **6. how the OOD detection performance works for EDL methods with $\lambda=0$?**
>
> We experimented with the $\lambda=0$ case, but observed that this simply encourages the model to output an extremely large value and leads to numerical errors during training. This is consistent with what Theorem 5.1 and Example 5.2 revealed, i.e., $\lambda=0$ corresponds to that the target for ID data is $\infty$.
>
> **7. What’s the benefits of three levels of uncertainty approach over Bayesian approaches?**
>
> First, our main claim in this paper is that distributional uncertainty alone while ignoring model uncertainty cannot capture uncertainty in a faithful manner. Second, we suggest that distillation-based methods could be a straightforward solution to remedy these issues. While distillation-based EDL methods have less computational complexity than Bayesian methods, we do not claim that distillation-based EDL methods are fundamentally superior, and further studies are warranted to compare the two approaches.
>
> **8. if the proposed method will be robust to learning deficiencies issue in EDL literature [1, 2]?**
>
> The learning deficiencies issue is mainly related to the activation function in the EDL model architecture, which might be a tangential problem to this work.
>
> **9. How does the aleatoric uncertainty behave with an increase in sample size?**
>
> Aleatoric uncertainty does not have asymptotic behavior. In practice, aleatoric uncertainty will converge to a constant when the model is trained with a sufficient number of samples.

---

> ### Comment · Reviewer_LBKP · 2024-08-10
> **Additional Clarifications on the Rebuttal**
>
> I thank the authors for the rebuttal response. However many of my concerns remain
>
> 2. It is unclear if there is any benefit of the bootstrap-based EDL framework over Bayesian approaches.
>
> Clarification regarding faster inference compared to ensembles
>
> My current understanding is that M different models would need to be trained on subset of training dataset to obtain the distribution p(\psi|D). I believe that the training is significantly more expensive than EDL approaches, and as expensive as ensemble of M models. I feel as if the inference would be as expensive as ensemble of M models. I wonder if my understanding is correct?
>
> 3. The proposed solution (bootstrap-Distil)'s aleatoric/epistemic uncertainty behavior is not well justified.
>
> a) The claim: The bootstrap-distill method does not have any hyper-parameters in its objective --> Wouldn't the bootstrap-Distil method introduce the hyperparameter: number of bootstrap samples?
>
> b) The clarifying question is: will it's uncertainties show reasonable trends? For eg. what is the trend of aleatoric/epistemic ucnertainty trend of bootstrap-Distil for 2D gaussian? I think for a model claiming fine-grained uncertainty quantification capabilities, both epistemic and aleatoric uncertainties should show meaningful trends. I may have missed but currently I'm not confident on the superiority of bootstrap-Distil's uncertainty (the claim: the aleatoric uncertainty is not model/hyper-parameter dependent, --> what is the uncertainty trend?)
>
> 9. How does the aleatoric uncertainty behave with an increase in sample size?
>
> Theoretically,  aleatoric uncertainty should converge to a constant when the model is trained with a sufficient number of samples. The clarifying question was regarding how the EDL and the proposed model's aleatoric uncertainty behaves with change in number of samples. I believe observing aleatoric uncertainty for the EDL model/proposed model across the experiments/datasets should be straightforward and reveal insights into model's uncertainty behavior.
>
> Unanswered claims:
>
>  Is the aleatoric/epistemic uncertainty trend of this model accurate on all datasets and settings or only accurate on simple dataset of Cifar10 (the epistemic uncertainty is shown only for one experiment of Cifar10) It is still not clear to me whether the proposed model's uncertainty be reasonable in realistic practical datasets (eg. imagenet/tiny-ImageNet/Cifar100)

---

> > ### Author Response · Authors · 2024-08-12
> >
> > We thank the reviewer for engaging in the discussion and providing additional comments. We would like to address the reviewer’s remaining concerns as follows.
> >
> > **1.Clarification regarding faster inference compared to ensembles.**
> > We acknowledge that the distillation-based method incurs a higher runtime compared to classical EDL methods, as mentioned in the last sentence of the abstract. This additional runtime primarily stems from the construction of $p(\psi|\mathcal{D})$, which often requires training multiple models or performing multiple inferences.
> >
> > However, the distillation-based method remains more computationally efficient than ensemble or Bayesian methods during inference at test time. For example, if $M$ ensemble models are trained on CIFAR-10 and distilled into a single EDL model, only this single EDL model is needed for OOD detection during inference. In contrast, the deep ensemble method would require storing and inferring all $M$ models.
> >
> > Overall, distillation-based EDL methods aim to emulate the desired properties of classical Bayesian or ensemble methods for faithfully quantifying uncertainties while being more computationally efficient (during inference).
> >
> > **2. Regarding hyper-parameters in its objective.**
> > We agree with the reviewer that the number of bootstrap samples $M$ can be viewed as a hyperparameter, i.e., using $M$ samples of $\psi$ to approximate the stochastic algorithm $p(\psi|\mathcal{D})$. We wish to remark, however, that larger $M$ would always lead to better approximation of $p(\psi|\mathcal{D})$, and thus a practitioner can choose the largest $M$ within the computational budget. This is qualitatively different from the hyperparameter $\lambda$ in the EDL methods, which has to be tuned based on a certain criterion with additional validation dataset.
> >
> > **3. Regarding both epistemic/aleatoric uncertainty trends of bootstrap distillation method.**
> > We appreciate the reviewer’s suggestion. We have conducted experiments per your suggestion and will include the trends of uncertainty measures with respect to sample size, across the different methods including the bootstrap distillation in the revision.
> > Specifically, we will update Figure 4(a) to include bootstrap distillation and add two additional figures like Figure 4(a) and Figure 10 to demonstrate the trend of aleatoric uncertainty with respect to the sample size.
> >
> > As we are not allowed to include a link or pdf file during the discussion period, here we provide a verbal description on the trend to be added in the revision.
> > - For epistemic uncertainty, the bootstrap distillation exhibits a vanishing trend as the sample size increases even for Gaussian data (which was missing in Figure 4(a)), as in Figure 10.
> > - For aleatoric uncertainty, the bootstrap distillation exhibits a decreasing trend as the sample size increases and converges to constant in both datasets (Gaussian and CIFAR10), as one can expect.
> >
> > **4. How does the aleatoric uncertainty behave as sample size increases?**
> > As alluded to above, a reasonable learned aleatoric uncertainty is expected to converge to constant as sample size increases. We plotted the aleatoric uncertainty trends for the same experiments setting in Figure 4 (Gaussian) and Figure 10 (CIFAR10), and found that other EDL methods also exhibit similar trends in general, except in a few cases. On Gaussian, all classical EDL methods and bootstrap distillation methods exhibit decreasing trend of aleatoric uncertainty as sample size increases, but classical EDL methods show some fluctuation after the aleatoric uncertainty converges to a certain level. On CIFAR10, all classical EDL methods and bootstrap distillation methods exhibit a monotonically decreasing trend of aleatoric uncertainty, except RPriorNet.
> >
> > The above observation justifies that main limitations of classical EDL methods are mainly twofold: (1) non-vanishing epistemic uncertainties and (2) hyper-parameter dependent aleatoric uncertainty. We will add a discussion on these new results in the revision, to clarify in which scenarios EDL methods exhibit expected or undesirable behaviors.
> >
> > **5. Results on larger scale datasets.**
> > We respectfully emphasize that the main focus of our submission is on the theoretical analyses of EDL methods to provide insights on their underlying principle and pitfalls, and illustrate the practical implications with synthetic and standard real data benchmarks.
> > We agree, however, with the reviewer that including results for more large-scale datasets would further strengthen our claims. We will incorporate more experimental results in the revision to address this.

---

> > > ### Comment · Reviewer_LBKP · 2024-08-13
> > > **Additional Comments**
> > >
> > > Thank you for the clarifications. After considering the rebuttal and author comments, many of my paper's understandings have been clarified.
> > >
> > > However, I still believe that the proposed model: distil-bootstrap to address the issues of EDL based works has significantly limited evaluation. The experiments and results can be improved significantly. Specifically, the uncertainty behavior (i.e. aleatoric/epistemic uncertainty) could be better studied. Both aleatoric and epistemic uncertainty decreasing with increase in sample size is a bit counter-intuitive to me. Ideally, aleatoric uncertainty should be reflective of the data noise
> > > and be constant with increase in sample size. Maybe in noisy data setting, the aleatoric uncertainty increases, and epistemic uncertainty remains constant. Currently, it is not clear if the uncertainty of developed model is reasonable.
> > >
> > > Also, experiments, and uncertainty analysis with more realistic datasets (Cifar100/tiny-Imagenet/ImageNet) can significantly strengthen the work.
> > >
> > > One minor thing after the rebuttal: I think the authors could include/discuss the distillation idea used in the paper (i.e. maybe discuss distillation used after lines 329-331) to better clarify that the p(\psi|D) is distilled to single EDL model. It was not obvious and a bit misleading to me from the first read of the draft.

---

> > > > ### Author Response · Authors · 2024-08-13
> > > >
> > > > We are glad that many points have been clarified with our response, and we wish to make a few final remarks.
> > > >
> > > > - We agree that the expected behavior of aleatoric uncertainty with increasing sample size is to converge to a constant, not necessarily monotonically decreasing. In our previous response, we also noted that `As alluded to above, a reasonable learned aleatoric uncertainty is expected to converge to constant as sample size increases.`, but we suspect that another sentence `On CIFAR10, all classical EDL methods and bootstrap distillation methods exhibit a monotonically decreasing trend of aleatoric uncertainty, except RPriorNet.` was misleading.
> > > >  To better demonstrate the behavior of the existing EDL methods and the bootstrap distillation, we will include an additional experiment on noisy data as suggested with a proper discussion.
> > > > - In the revision, we will certainly include a result on larger datasets like Cifar100/tiny-Imagenet/ImageNet, as suggested.
> > > > - We will revise lines 329-331 to clarify the bootstrap distillation method to avoid any further confusion.
> > > >
> > > > We once again appreciate the reviewer's engagement in the discussion.

---

### Official Review · Reviewer_Sxes · 2024-07-13

**Soundness:** 3
**Presentation:** 3
**Contribution:** 3
**Rating:** 7
**Confidence:** 4

**Summary:**

In this paper, the authors propose a novel analysis of existing evidential learning approaches, which have recently gained significant attention in the domains of uncertainty estimation and probabilistic modeling. The two major contributions of this work are as follows: First, the authors provide a clearer understanding of the asymptotic behavior of a wide class of EDL methods by unifying various objective functions. Second, they reveal that EDL methods can be better interpreted as out-of-distribution detection algorithms based on energy-based models. Specifically, the first contribution explains why evidential learning might have poor uncertainty quantification capabilities (or probabilistic modeling quality), while the second explains why EDL methods empirically show good performance on a number of downstream tasks. The authors validate their insights through extensive experiments on several image classification tasks (CIFARs and TinyImageNet) and EDL baselines.

**Strengths:**

* The paper is clearly written and easy to follow. The idea is intuitive and easy to grasp. The related work section provides an adequate discussion of existing approaches to anchoring-based training. The analysis narrative, with the presented drawbacks of existing methods, is very clear and easy to understand.

* The authors derive a novel way to unify existing evidential learning approaches within Unified EDL Objectives, which provides a new perspective on these types of models and facilitates further analysis. As an example, the authors demonstrate that a typical EDL approach suffers from asymptotic behavior when aleatoric uncertainty does not vanish with an increasing training set and remains constant for ID data.

* The analysis also provides an interesting finding that the EDL models could be considered within the energy-based OOD framework, which partially explains their success in a number of downstream tasks.

**Weaknesses:**

* From the perspective of the experimental evaluation, I would be curious to see evidence that the behavior demonstrated in the paper would hold in other domains, such as texts, graphs, more complicated vision tasks (e.g. segmentation), not limiting to image classification task. Additionally, evidential learning also covers regression tasks, which would also be very beneficial for the paper to discuss.

* The authors conducted their evaluation exclusively using ResNet18 and VGG16 models. Incorporating more recent models and state-of-the-art architectures would likely provide a more comprehensive and robust assessment of their approach, ensuring validity of the results across a wider range of scenarios.

The paper does not have any major flaws or weaknesses that would warrant rejection, and I tend to assess the paper positively. I think that the paper would be very interesting for the community since it provides several important insights about a popular family of uncertainty estimation approaches.

**Questions:**

* How do the proposed results translate to the regression case for evidential learning? Is there any way to extend the mentioned insights into this case? For example, it seems that the EBM intuition of OOD provided for classification potentially wouldn't work for the regression case.

* The paper highlights a very interesting problem with EDL models, which have gained a lot of attention recently. In Section 6, the authors discuss the reason for these problems and the potential solution, which is stochastic modeling either through ensembling or dropout (plus the proposed bootstrapping). If stochasticity is a solution for EDL drawbacks, could we apply efficient ensembling alternatives [1, 2, 3, 4] to achieve the same results? I think it could be valuable to discuss this.

[1] Wen, Yeming, Dustin Tran, and Jimmy Ba. "Batchensemble: an alternative approach to efficient ensemble and lifelong learning." ICLR 2020

[2] Durasov, Nikita, et al. "Masksembles for uncertainty estimation." CVPR 2021

[3] Laurent, Olivier, et al. "Packed-ensembles for efficient uncertainty estimation." ICLR 2023

[4] Turkoglu, Mehmet Ozgur, et al. "Film-ensemble: Probabilistic deep learning via feature-wise linear modulation." NeurIPS 2022

---

> ### Author Rebuttal · Authors · 2024-08-07
>
> We thank the reviewer for their thoughtful comments. We address them in detail below.
>
> **1. Whether EDL’s behavior demonstrated in the paper would hold in other modalities and domains.**
>
> We totally agree with the reviewer that analyzing EDL’s behavior in other domains and modalities is worth exploring. Since the classification setting for the vision domain is standard in the UQ literature, however, we focus on this setup to clearly demonstrate the implication of our analyses. While we leave such extensions as future work, we refer the reviewer to Appendix F.2, where we provide some analysis for the regression case.
>
> **2. Incorporating more recent models and state-of-the-art architectures for evaluation.**
>
> This is also a good suggestion. In fact, most EDL methods evaluate their performance on classical neural networks and standard benchmark datasets. Conducting a comprehensive empirical study to benchmark all existing EDL methods under a unified framework would also be an interesting future work.
>
> **3. How do the proposed results translate to the regression case? The EBM intuition of OOD wouldn't work for the regression case.**
>
> As the reviewer noted, the EBM intuition would not hold for the regression case. However, a similar observation can be made: the existing EDL methods for regression only aim to fit to a “fixed” uncertainty target. This is consistent with the observation made by a recent paper (Meinert et al., 2023). We kindly refer the reviewer to Appendix F.2, where we elaborate further on this case.
>
> **4. If stochasticity is a solution for EDL drawbacks, could we apply efficient ensembling alternatives to achieve the same results?**
>
> In this work, we argue that the EDL framework can be better used as a computational  tool to emulate the behavior of classical ensemble approaches, while reducing their computational burden. In this context, using more advanced ensembling techniques can indeed boost the UQ performance of distillation-based EDL methods. We will add the relevant discussion in the revision.

---

> > ### Comment · Reviewer_Sxes · 2024-08-12
> >
> > Thank you for your thoughtful engagement in the rebuttal. I appreciate the authors' efforts in addressing my concerns and providing additional insights, particularly in discussing the potential extensions of their work to other domains and modalities. The clarification on the regression case and the consideration of efficient ensembling alternatives also add valuable context to your findings. While the paper focuses on standard classification settings within the vision domain, the discussions provided indicate promising directions for future work. Given these improvements and the solid contributions made, I am increasing my original score by 1 point.

---

> > > ### Author Response · Authors · 2024-08-12
> > >
> > > We thank the reviewer for the positive feedback and increased score. We also appreciate the insightful comments, which have given us valuable ideas for potential extensions of this work.

---

### Author Rebuttal · Authors · 2024-08-07

# To ALL Reviewers:
We thank all the reviewers’ effort in reviewing our paper and providing thoughtful comments. We would like to take this opportunity to further clarify our contribution, and resolve some of the common concerns as follows:

**1. Theoretical justification of proposed Bootstrap Distillation**

As several reviewers asked about a further justification of the proposed bootstrap distillation in Section 6, we provide our response here.

- We first emphasize that the main contribution of this paper is to analyze the common limitation of EDL methods with empirical support, show that standard EDL methods simply fit the UQ model to a fixed target (Section 5.1), and reveal that the learned uncertainties bear no statistical meaning. Since this is due to the ignorance of model uncertainty in the framework, we show that incorporating model uncertainty in the EDL framework via distillation can alleviate the characterized issues at the cost of computational complexity. We propose the bootstrap procedure in this context as a new mechanism to induce model uncertainty. While we do not claim that bootstrap distillation (BD) is the best approach, our experiment shows that BD attains a good downstream task performance even compared to the existing distillation based approaches END2 and S2D; see Tables 2, 3, and 4 in Appendix.
- This suggests that a further study of the distillation-based EDL framework is warranted. We acknowledge that the BD lacks a rigorous justification, but conducting a comprehensive theoretical analysis of the method is rather nontrivial. As reviewer gUpP also suggested, such an analysis could be considered as a separate publication in the future.
- While it is relatively easy to argue why the epistemic uncertainty would vanish in the sample limit ($|\mathcal{D}|\to\infty$) with the bootstrap from the intuition (Section 6), we believe that a more sophisticated asymptotic analysis for vanishing epistemic uncertainty can be carried out with overparameterized neural networks, adopting a similar setting in [A]. Succinctly speaking, if the trained model's prediction can be shown to be asymptotically normal in the limit of the sample size, we can argue that the uncertainty captured by bootstrap behaves as Gaussian of vanishing variance in the sample limit. This implies a naturally vanishing (epistemic) uncertainty. We will carry out the analysis and discuss this in more detail in the revision.

[A] Huang et al., Efficient Uncertainty Quantification and Reduction for Over-Parameterized Neural Networks, NeurIPS 2023.

**2. New experiment Results**

As some reviewers suggested, we provide additional experimental results to further strengthen our work. We summarize the results below and the accompanying figures can be found in the attached pdf:

- (1) **Calibration Performance.** We measure and compare the Expected Calibration Error (ECE) score of different EDL methods in Figure 13. Calibration measures how well the model's confidence aligns with its prediction accuracy. Good calibration performance also indicates accurate “total uncertainty” quantification. From Figure 13, we observe a behavior similar to the selective classification task in Figure 12. Distillation-based methods achieve lower ECE scores on calibration and better AUROC scores on the selective classification task, which relies on total uncertainty. We also observe that Fisher-EDL achieves the lowest calibration error, aligning with its promising performance on the selective classification task in Figure 12.
- (2) **Aleatoric Uncertainty Benchmark.** In Section 5.2 and Section H.2 of Appendix, we argue that using a Dirichlet-framework-independent objective that promotes the same behavior as the EBM-based OOD detection algorithm suffices for downstream task performance. We justify this through the epistemic uncertainty benchmark. We further examine the behavior of aleatoric uncertainty quantified by EDL methods. First, we provide a visualization of aleatoric uncertainty on 2D Gaussian data in Figure 14. Second, we evaluate EDL methods’ capability to detect ambiguous data (linear interpolation between two test images) using aleatoric uncertainty in Figure 15. The results in Figures 14 and 15, together with Section 5.2, further support that EDL's UQ capability (on both epistemic and aleatoric uncertainty) is not due to the Dirichlet framework but rather due to other auxiliary techniques.
- (3) **Comparison with a cross-entropy-trained deterministic network.** Due to space limitations, we summarize the results in words as follows. On the OOD detection task, the cross-entropy-trained network achieves an average AUROC score of 91.6 (66.8) on CIFAR10 (CIFAR100). On the selective classification task, the cross-entropy-trained network achieves AUROC scores of 90.2 (84.1) on CIFAR10 (CIFAR100). The performance of this baseline is suboptimal compared to most EDL methods, especially on the OOD detection task, as expected. We will add the concrete results (using tables and figures) in the revision.
- (4) **Behavior of deep ensemble’s epistemic uncertainty.** Although the behavior of classical Bayesian approaches and ensemble approaches has been well studied in the literature, we provide the epistemic uncertainty vs. sample size curve of the deep ensemble method in Figure 16. Unlike EDL methods, the epistemic uncertainty of the deep ensemble demonstrates a vanishing trend, as expected.

---

### Author Response · Authors · 2024-08-07
**Main takeaway of this work**

To better emphasize our main points, we provide a quick overview of the paper with key takeaways:

- EDL methods have gained significant attention in the UQ community due to their empirical successes in downstream UQ tasks and computational efficiency. Despite these practical benefits, we believe that a more fundamental understanding is warranted to build a more reliable machine learning system based on the developed techniques. Hence, we aim to analyze how (most of) EDL methods learn uncertainty and its practical implications.
- As the first step, we unify several different training objectives of representative UQ methods in the literature. While they were motivated and derived from different motivations, and were thus considered to be different objectives, our framework provides a single framework to analyze the behavior of the existing EDL methods (Section 4). This allows a more concise and widely applicable analysis of representative existing EDL methods, especially compared to (Bengs et al., NeurIPS 2022).
- Second, while several EDL methods have been justified using the Bayesian framework, such as variational inference, we theoretically and empirically uncover that EDL methods behave quite differently from classical Bayesian methods in terms of both aleatoric and epistemic uncertainty (Section 5.1). This raises fundamental questions about the meaning of the learned uncertainties  of EDL methods.
- Third, we argue that EDL methods can be better understood as EBM-based OOD detection algorithms, highlighting several similarities between the two seemingly different approaches (Section 5.2). This provides a new perspective on why the EDL methods behave differently from the classical Bayesian approaches.
- Fourth, we empirically show that the claimed successes of several EDL methods for UQ downstream tasks are quite susceptible to several factors of their auxiliary techniques in the framework (Section 5.3).
Finally, we identify that the aforementioned issues with EDL are due to ignoring the model uncertainty for computational efficiency, and argue that the distillation-based methods could potentially be a better solution to remedy the issues, at the cost of the computational inefficiency (Section 6). This suggests an inherent trade-off between computational efficiency and statistical faithfulness in the framework.

Overall, this work aims to call the community’s attention to carefully re-examine the EDL approach. For practitioners, EDL methods could still be utilized as efficient algorithms for certain applications, such as OOD data detection. However, since the EDL approach was originally proposed as a tool to learn and distinguish epistemic uncertainty and aleatoric uncertainty, a more thorough understanding is required for its appropriate usage in future applications towards building more reliable machine learning agents. We believe that our work provides a useful perspective in this direction.

---

### Decision · Program_Chairs · 2024-09-25

**Decision:**

Accept (poster)

**Comment:**

To all reviewers I would like to thank you for the time and effort you have put into the review process and for the in depth discussion with the authors. To the authors, thank you so much for engaging with the reviewers and for submitting your paper to NeurIPS.

After reading all of the reviews and all of the rebuttals, and reading the paper itself in depth, I would recommend this paper to be accepted.
I think this is a novel method to look at aspects of quantify Uncertainty in such models. This is heightened by the analytical treatment of the problem at hand and  specifically I think Lemma 5.1 can be something the community build upon and expand perhaps by testing with more empirical experiments. I think this is enough to compensate for the small scale fo experiments.

I would like to highly encourage the Authors to take into account all of the feedback  given and incorporate the remarks for the camera ready copy.

Best wishes,

    AC